# Is transport of microplastics different from mineral particles? Idealized wind tunnel studies on polyethylene microspheres.

Eike Maximilian Esders[1], Sebastian Sittl[2], Inka Krammel[2], Wolfgang Babel[1,3], Georg Papastavrou[2], and Christoph Karl Thomas[1,3]

[1]Micrometeorology Group, University of Bayreuth, Universitätsstraße 30, Bayreuth, Germany
[2]Department of Physical Chemistry II, Universtiy of Bayreuth, Universitätsstraße 30, Bayreuth, Germany
[3]Bayreuth Center of Ecology and Environmental Research, Dr.Hans-Frisch-Str.1-3, Bayreuth, Germany

**Correspondence:** Eike Esders (Eike.Esders@uni-bayreuth.de)

**Abstract.** Atmospheric transport can disperse microplastic particulate matter to virtually every environment on the planet. Only few studies have examined the fundamental transport mechanisms for microplastics and contrasted them with the existing body of knowledge accumulated for mineral dust over the past decades. Our study addresses this research gap and presents results from idealized wind tunnel experiments, which examine the detachment behavior of microplastics ranging from 38 to 125 $\mu$m in diameter from smooth substrates. We here define 'detachment' as microspheres detaching from a substrate and leaving the field of observation, which includes several transport modes including creeping, rolling, directly lifting off. The detachment behavior of polyethylene microspheres (PE69) and borosilicate microspheres (GL69) of nominally the same physical diameter (63-75 $\mu$m) are contrasted across hydrophilic to hydrophobic substrates. We further examine the effect of microsphere-microsphere collisions on the detachment behavior of both polyethylene and borosilicate microspheres. In a collision the rolling micropshere can detach a static microsphere or be stopped by it. Differentiating between microspheres experiencing only fluid forces and microspheres experiencing fluid forces and collisions, revealed that collisions can faciliate and mitigate detachment. Further, results indicate that GL69, as a hydrophilic particle, is sensitive to substrate hydrophobicity whereas PE69 is not sensitive. When sensitive, microspheres detached more easily from hydrophobic substrates compared to hydrophilic substrates. The smallest polyethylene microspheres behave similar to borosilicate microspheres. Results demonstrate that PE69 and GL69 as proxy for plastic and mineral dust, respectively, detach at $u_*$ between 0.1 to 0.3 ms$^{-1}$ fitting the prediction of a fluid treshold model by Shao and Lu (2000). In the observed range of rH, capillary forces can increase the median detachment about 0.2 ms$^{-1}$ for PE69 and GL69. The smallest polyethylene microspheres, behaved similar to borosilicate microspheres, by being sensitive to the substrates hydrophobicity. For bigger microspheres, the lesser density of polyethylene drives their higher erodibility. At similar relative humidity polyethylene microsphere detach at smaller friction velocities compared to borosilicate microspheres of the same nominal diameter. We argue that our idealized experiments provide a useful analog to more complex experiments, for example using simple soils as substrate. We conclude that plastic particles are preferentially transported, as their lower density and more hydrophobic surface facilitate detachment.

# 1  Introduction

Humans use plastics in virtually all their activities with the consequence of bringing around 4900 Mt of plastic waste into the environment from 1950 to 2015 (Geyer et al., 2017). Plastics are intentionally non-biodegradable and can persist in the environment for several centuries (Barnes et al., 2009). Exposed to the environment, plastic negatively affects individual organisms (Alexiadou et al., 2019; Donnelly-Greenan et al., 2019) and ecosystems (Windsor et al., 2019; Wang et al., 2022; Zhang et al., 2022).

The present work focuses on microplastic, defined as plastic particles smaller than 5 mm in all dimensions. Based on their origin, microplastics are further subdivided into either primary microplastics, which are produced with sizes $\leq$ 5 mm, or secondary microplastics formed by degradation of larger plastic particles (Weinstein et al., 2016; Du et al., 2021; Meides et al., 2021).

Today, microplastics are found in animals (Carlin et al., 2020; Ugwu et al., 2021; Thrift et al., 2022), terrestrial systems (Xu et al., 2020; Chia et al., 2021), the atmosphere (Zhang, 2020; Allen et al., 2021) and in aquatic systems (Thushari and Senevirathna, 2020). Further, recent evidence shows microplastics move between aquatic, atmospheric and terrestrial systems (Horton and Dixon, 2018; Bank and Hansson, 2019; Brahney et al., 2020; Evangeliou et al., 2020; Lehmann et al., 2021; Rehm et al., 2021; Boos et al., 2021; Kernchen et al., 2022; Rolf et al., 2022; Shiu et al., 2022).

Compared to aquatic systems, microplastics transport and concentrations in terrestrial and atmospheric ecosystems have gained much less attention (Li et al., 2020). Strikingly, it is likely that arable soils alone contain more microplastics than the oceans (Rillig, 2012; Nizzetto et al., 2016; de Souza Machado et al., 2018). Further, arable soils are susceptible to wind erosion and thus potentially source areas for the atmospheric transport of microplastics (Rezaei et al., 2019; Tian et al., 2022; Yang et al., 2022). Regarding atmospheric transport of microplastics, most research has focused on atmospheric deposition (Zhang, 2020). Deposition was measured in cities (Dris et al., 2015; Klein and Fischer, 2019; Shruti et al., 2022) with typical deposition rates ranging between $\leq$ 5 and $\geq$ 1000 microplastics$\cdot m^{-2} d^{-1}$. Further, microplastics were found in remote environments such as the arctic, nature reserves, glaciers and the deep sea (Katija et al., 2017; Allen et al., 2019; Bergmann et al., 2019; Brahney et al., 2020; Stefánsson et al., 2021; Materić et al., 2022). Their distance to urban areas suggests that microplastics are transported via the atmosphere rather than via aquatic systems. So far, little is known about microplastic transport and dynamics in the atmosphere (Allen et al., 2022).

The atmospheric transport of microplastics starts with the detachment from a substrate. This process has only recently been studied. The saltation of sand particles is a key mechanism for sand transport and dust emission Gillette (1981). Dust particles experience strong cohesive forces Iversen and White (1982). Aerodynamic drag acting on dust particles is usually not strong enough to break the interparticle bonds. However, impacting sand particles easily disrupt the interparticle bonds and eject dust particles. The following studies examined plastic particle emission driven by sand saltation. Tian et al. (2022) and Yang et al. (2022) presented studies conducted on arable land in northern China, which used novel flat open traps to collect in-situ wind-blown sand (saltation) and dust (suspension) during wind erosion events. Bullard et al. (2021) presented a series of wind tunnel experiments with prepared mixtures of soil, sand, and microplastics. Rezaei et al. (2019) used a movable wind tunnel to

conduct controlled erosion experiments on arable and natural environments. The mentioned studies yielded enrichment ratios of microplastics and relate the concentration of microplastics in the original substrate to that in the transported soil. All findings showed that microplastics are preferentially transported by wind in comparison to the mineral soil. Typical enrichment ratios were found to be as high as 16.6 for fragments, and up to 726 for fibers. The relative lower density of any microplastic and the elongated form of microplastic fibers are hypothesized to drive the preferential transport resulting in the high enrichment ratios.

The presented explorative studies are the first step in uncovering the mechanisms driving the emission of microplastics, which requires investigating the fundamental mechanisms of their movement in the environment. As a start, the present work investigates the detachment of a monolayer of factory-fresh polyethylene microspheres and borosilicate microspheres from smooth substrates with hydrophilic to hydrophobic surfaces in a series of wind tunnel experiments. By studying the detachment behavior of the microspheres, the influence of material properties and relative humidity (rH) on the detachability of both microsphere types will be elucidated.

A microsphere on a substrate is influenced by gravity, adhesion, aerodynamic drag and aerodynamic lift. Gravity and adhesion are the stabilizing forces. For microspheres smaller than 50 $\mu$m, the force of adhesion is at least 100 times larger than the gravitational force (Shao and Lu, 2000), which becomes relevant for microspheres larger than 100 $\mu$m in size. Further, adhesion has a non-linear relation to rH. At some critical relative humidity (rH$_c$), capillary forces occur, due to water accumulating between microsphere and substrate. When capillary forces are present, adhesion is increased, but when capillary forces are not present, adhesion is constant and independent of rH (Corn and Stein, 1965; Ibrahim et al., 2004; Rabinovich et al., 2002; Kim et al., 2016). Thus, rH must be known to interpret the detachment of microspheres from a substrate. At best, laboratory studies control rH and directly compute rH$_c$. In the presented work, rH was monitored during the experiments, but controlling the moisture content of the air was outside the scope and technical possibilities for airflows of several thousand $m^3 h^{-1}$. Nonetheless, natural, weather-driven variations of rH during the experimental phase allowed us to evaluate a wide range of relative humidities since the air entering the wind tunnel communicated with outside conditions.

Aerodynamic drag and aerodynamic lift are the detaching forces. These two aerodynamic forces are related to the shear forces close to the surface, and hence are functions of the density-normalized momentum flux, termed the friction velocity ($u_*$). When the detaching forces overcome the stabilizing forces, a microsphere can be detached from the substrate it rests upon. Regarding soils, Kok et al. (2012) defines the friction velocity at which particles start to move solely by fluid forces as the fluid threshold ($u_{*,ft}$). In analogy to the fluid threshold, we define an idealized fluid threshold ($u_{*,ift}$). Idealized, as in the presented study, microspheres are dispersed as a monolayer on a smooth substrate.

When a microsphere detaches from a smooth substrate, it can roll, slide or lift off directly (Kassab et al., 2013). The idealized fluid threshold increases from rolling to sliding to directly lifting-off (Soltani and Ahmadi, 1994; Ibrahim et al., 2004). Note that, on a rough substrate it is more likely, that microspheres lift off directly (Kassab et al., 2013). However, due to the stated findings, we assume that microspheres roll on the substrate. As the microspheres roll, collisions between moving microspheres and stationary microspheres are likely. Ibrahim et al. (2004) showed that collisions effectively detach microspheres and that moving microspheres can transfer more kinetic energy onto a stationary microsphere during collision compared to the acting

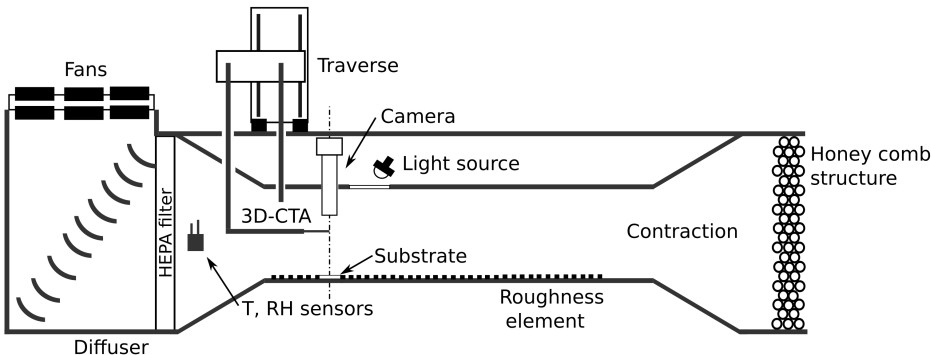

**Figure 1.** Schematic of the wind tunnel to observe microsphere detachment.

aerodynamic forces. Thus, we define a collision threshold $u_{*,ct}$. It is smaller than the fluid threshold $u_{*,ct} < u_{*,ift}$, if collisions are effective. Similarly, a moving microsphere can be stopped by a stationary microsphere, when the collision impulse is insufficient to overcome the binding forces.

The main motivation of the work presented here is driven by the questions i) to what extent do size, surface properties, and relative humidity influence the idealized fluid threshold and collision threshold of PE and borosilicate microspheres? And ii) to what extent do the findings support the preferential transport of microplastics found in former studies?

## 2    Methods

### 2.1    Wind tunnel and camera equipment

The non-circulating wind tunnel used in this study was 730 cm long, 60 cm wide and 120 cm high (see Fig. 1). Air is sucked sequentially through a contraction section (cross-section dimensions of 54 cm x 27 cm; contraction ratio 5:1; honeycomb structure ensuring laminar flow conditions), a test section (substrate mount, camera, constant temperature anemometer and light source), and a deflector section (HEPA filter; temperature and relative humidity sensors). The outlet held twelve fans of diameter 26.5 cm (RAB O TURBO 250, DALAP GmbH, Germany), whose rotation speed was controlled using a stepless

transformer (LSS 720-K, Thalheimer Transformatoren GmbH, Germany). The rotation speed of the fans controlled the airflow speed in the wind tunnel. The air was filtered by a HEPA filter (EU2, 10 $\mu$m pore size, Erwin Telle GmbH, Germany), before it was released to the laboratory.

A roughness element, installed for turbulence production, covered most of the test section's floor (170 cm). The roughness element was equipped with a substrate mount such that its surface containing the microspheres was at the same vertical height

as the top of the roughness element. The background of the substrate mount was painted with a high emissivity resistant black paint for high contrast between microspheres and image background (HERP-LT-MWIR-BK-11, LabIR, Czech Republic). Above the substrate mount a constant temperature anemometer (CTA) (three-dimensional hot wire probe, Model 55P095,

Dantec Dynamics, controller, Model 54T42, Dantec Dynamics) measured the turbulent flow statistics. The deflector section redirected the air to the outlet of the wind tunnel. In the deflector section, air temperature and rH were measured using a slow-response thermohygrometer (Model HC2A, rotronic). In addition, the CTA was equipped with a fast-response air thermometer (Model 90P10, Dantec Dynamics) located in proximity to the test substrates.

The free-stream velocity could be varied from zero up to 11 ms$^{-1}$. It was measured at half the height of the wind tunnel at z = 270 mm. The free stream turbulence intensity ($I$), was less than 1 % over the entire velocity range. It is defined as:

$$I = \sigma_u / \bar{U} \tag{1}$$

where $u$ represents instantaneous horizontal wind velocities, $\sigma_u$ is the standard deviation of $u$ and $\bar{U}$ is the horizontal mean velocity.

Microsphere detachment was captured with a camera (Model Sony Alpha 7 RII, Sony) equipped with a long-distance microscopy lens (K2 DistaMax, with a CF1 lens, Infinity, USA). The long-distance microscope lens captured images with a magnification ratio of up to 1:1 at an object distance of 35 cm. The photo equipment did not disturb the flow field due to its large distance to the substrate. The field of observation was 861.6 mm$^2$. Images were taken every 10 seconds and were directly transferred and stored on a PC. Experiments were conducted at an air temperature of 23 $\pm$ 2 °C and an air relative humidity from 20 % to 60 %.

## 2.2 Turbulence characteristics

The vertical velocity profiles showed a typical boundary-layer velocity profile for a channel flow (see Fig. A3). The friction velocity and roughness length were calculated for z $\leq$ 21 mm, where the velocity profile agrees well with the logarithmic law of the wall. The roughness length ($z_0$) was calculated by extrapolating the logarithmic wind profile:

$$\bar{U}(z) = \frac{u_*}{\kappa} ln \left( \frac{z}{z_0} \right) \tag{2}$$

to the height z where $\bar{U}$ = 0, $\kappa$ is the Karmann constant with a value of 0.4, giving $z_0$ = 0.5 mm.

The friction velocity ($u_*$) was computed against the free-stream velocity ($U_\infty$). The velocity was regressed by a least-squares linear algorithm to

$$u_* = 0.06 \cdot U_\infty, \tag{3}$$

where the uncertainty in $u_*$ is 0.02 ms$^{-1}$ for the 99.7 % percentile.

## 2.3 Determining detachment from measurements

Detachment of microspheres was measured as the decreasing number of microspheres in the field of observation. The algorithm used to quantitatively determine the number of microspheres in each individual image is described in Esders et al. (2022).

| Substrate | $\Theta_S$ (°) | Substrate material | Cleaning procedure | Coating |
|-----------|----------------|--------------------|--------------------|---------|
| A | <30 | glass plate | RCA | no coating |
| B | $55 \pm 5$ | glass plate | Ibrahim et al. 2003 | no coating |
| C | $65 \pm 1$ | glass plate | RCA | 3-aminopropyldimethylethoxysilane |
| D | $120 \pm 1$ | glass plate | RCA | 1H-1H-2H-2H perfluorodecyltrichlorosilane |

**Table 1.** Overview of the four types of substrates used in the experiments. The static contact angles ($\Theta_S$) of substrates with a water droplet indicate hydrophilicity or hydrophobicity. A small $\Theta_S$ indicates a hydrophilic, a high $\Theta_S$ indicates a hydrophobic substrate. Substrates were cleaned according to a protocol developed by the radio corporation of america (RCA) Kern (1990) or a cleaning procedure described in Ibrahim et al. (2003). Substrates c and d were coated with 3-aminopropyldimethylethoxysilane and 1H-1H-2H-2H perfluorodecyltrichlorosilane, respectively, after the cleaning procedure.

The algorithm labels the fluorescent particles and gives the total number of particles for every image. With increasing wind speeds, the number of particles decrease as they are transported out of the field of observation. In a single experiment, up to approximately 1500 microspheres were placed on a substrate as described above, positioned in the wind tunnel on the substrate mount, and subsequently exposed to airflows with increasing friction velocities. The friction velocity ($u_*$) increased step-wise from 0 to 0.65 ms$^{-1}$ while it was held constant for 360 s at each step. The increments in $u_*$ were selected such that at every step more than 10 % of the initial number of microspheres, but less than 30 % were detached to optimize the statistics on moving microspheres. A single experiment comprised about 200 images.

### 2.4 Microspheres and substrates

We used polyethylene and borosilicate microspheres to address our research questions. Three differently sized spherical fluorescent plastic microspheres made of polyethylene (density: 1025 kgm$^{-3}$; Cospheric LLC, United States) were used with the following diameters: 38-45 $\mu$m (hereafter referred to as PE42), 63-75 $\mu$m (hereafter referred to as PE69), and 106-125 $\mu$m (hereafter referred to as PE115). The borosilicate microspheres with diameters 63-75 $\mu$m were used (density: 2200 kgm$^{-3}$; Cospheric LLC, United States), hereafter referred to as GL69.

Comparing PE69 and GL69, is especially interesting, as they have the nominal same diameter. We chose them as the directly compared proxies for plastic and mineral dust. For PE69 and GL69, the root-mean-square roughness was determined, being 248.5 $\pm$ 32.2 nm and 27.7 $\pm$ 9.0 nm, respectively (see Fig. A7 and A8). See section A6 for a description of the surface roughness measurement technique. Further, see section A5 for a scanning electron images of PE69 and GL69.

The microspheres were detached from glass plates as substrate material (76×26 mm, Thermo Scientific; 76×26 mm, VWR). Substrates with hydrophobic to hydrophilic surfaces were prepared by cleaning and subsequent optional coating (see Tab. 1). The hydrophobicity of a substrate was characterized by its static contact angle with a water droplet ($\Theta_S$) using the sessile drop method (Dataphysics, Contact Angle System OCA, Filderstadt, Germany). A substrate with a small $\Theta_S$ is hydrophilic and with a high $\Theta_S$ is hydrophobic. A detailed preparation procedure for each substrate is described in the supplemental information.

| Microsphere | Number of measurements | | | | Initial number of microspheres | Mean number density [$\#/mm^2$] |
| | *Substrate A* | *Substrate B* | *Substrate C* | *Substrate D* | | |
| --- | --- | --- | --- | --- | --- | --- |
| PE42 | 9 | 9 | 9 | 2 | $1490 \pm 950$ | 0.9 |
| PE69 | 11 | 9 | 9 | 6 | $647 \pm 490$ | 0.3 |
| GL69 | 16 | 9 | 9 | 9 | $366 \pm 165$ | 0.5 |
| PE115 | 11 | 9 | 5 | 9 | $1164 \pm 514$ | 0.2 |

**Table 2.** Overview of how many experiments were conducted regarding every microsphere - substrate combination. The initial number of microspheres indicates, how many microspheres were observed at the beginning of an experiment. The mean number density indicates, how many microspheres are in mm$^2$ on average.

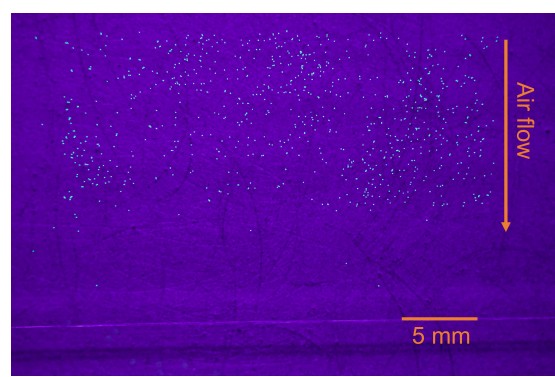

**Figure 2.** Typical substrate just before an experiment, here equipped with polyethylene microspheres with 63 $\mu$m to 75 $\mu$m in diameter. In respect to the image, air flows from top to bottom.

The microspheres were deposited on the substrates via the sealing cap of their containers. The closed microsphere container was flipped once, thus a few microspheres would adhere to the sealing cap. Then the sealing cap was screwed off the container and by tipping on the top of it, small amounts of microspheres could be released onto the substrate. Substrates were prepared outside the wind tunnel. Deposition was made immediately before the start of an experiment to minimize the residence time of the microspheres, as residence time can suppress detachment (Ibrahim et al., 2004). A self-made template (see Fig. A1) for microspheres deposition, ensured that all microspheres were deposited in the field of observation (see Fig. 2). For information on performed experiments, see Table. 2.

A UV-light (LED SLS-6 UV Floor, eurolite, Germany) was used to excite fluorescent microspheres, while a daylight lamp (Lumilux Cool Daylight, L18W/865, Osram, Germany) was used to expose the borosilicate microspheres.

### 2.5 Determining the effect of collisions on detachment

Collisions between microspheres were likely as initial number densities of deposited microspheres reached up to 1.7 microsphere $\cdot$ mm$^{-2}$ and certainly impacted the detachment statistics reported below. Two cohorts of microspheres were defined to

study the effect of collisions on detachment. The first cohort excluded the effect of collisions on detachment, while the second cohort included the effect of collisions on detachment specifically. For the first cohort, the most windward microspheres in the field of observation in the first image were determined. For each subsequent image, the decrease in these particles was registered. The results of this analysis give the idealized fluid threshold for a single experiment. The algorithm is described in detail in the appendix. Note that, the most windward microspheres cannot be impacted by downwind moving microspheres. For the second cohort, all microspheres in the field of observation were counted, but the remaining most windward microspheres were subtracted. The results of this analysis give the collision threshold.

## 2.6 Characteristic microsphere statistical quantities

The results relate the detached fraction to the friction velocity of the turbulent airflow. This fraction is defined as,

$$N_*(u_*) = 1 - \frac{N(u_*)}{N(0)} \tag{4}$$

where $N(u_*)$ is the number of non-detached microspheres on the substrate, after it was exposed to a friction velocity $u_*$ and $N(0)$ is the initial number of microspheres on the substrate. A logistic function of the following form was fit to the results of individual experiments,

$$N_*(u_*) = \frac{A}{1 + e^{-b \cdot (u_* - \frac{A}{2})}} \tag{5}$$

where $A$ is the function's maximum value and $b$ is the logistic growth rate. The fit allowed to evaluate $N_*(u_*)$ at any $u_*$ which allowed for comparison between individual runs. Further, the relative number of detached microspheres is defined as:

$$N_*(t) = 1 - \frac{N(t)}{N(0)} \tag{6}$$

where $N(t)$ is the number of non-detached microspheres on the substrate at time t and $N(0)$ is the initial number of microspheres on the substrate (t = 0). For the respective cohorts, the idealized fluid threshold ($u_{*,ift}$) and the collision threshold ($u_{*,ct}$) were determined as the value at which 50 % of all microspheres observed in an experiment detach. We use this value to represent the overall detachment behavior.

Note that a population of microspheres detaches over a range of velocities rather than at a specific single velocity. Every microsphere experiences an individual force of adhesion depending on its exact diameter and its surface roughness in relation to that of the substrate, and hence has an individual critical friction velocity. Further, the stochastic nature of the turbulent airflow exerts a homogenous force onto the substrate when averaged over longer time scales, while on short time scales its substantial spatio-temporal variability is large.

To summarize, we represent the idealized fluid threshold or collision threshold of a single experiment by a single value, a friction velocity. There are always microsphere detaching at higher or smaller velocities, due to the stochastic nature of adhesive forces and turbulence. See figure A9 for examples of the fit of the logistic model to experimental data.

## 2.7 Predicting the fluid treshold

Shao and Lu (2000) proposed a model, that predicts the fluid threshold for a soil that is made up of uniform, spherical microspheres that are spread loosely over a dry and flat substrate. Hereafter, the model is referred to as the Shao model. We propose that a glass plate as a substrate equipped with a monolayer of microspheres is an analogy to a simplified soil. Thus, we will compare the results to the prediction of the Shao model. The model was used in the following form:

$$u_{*,ft} = A_N \sqrt{\sigma_\rho g d + \frac{\gamma}{\rho d}} \tag{7}$$

where $A_N = 0.111$, $\sigma_\rho$ is the ratio of the microsphere's density to the density of air, $g$ is the gravitational acceleration, $d$ is the diameter and $\gamma$ is $3 * 10^{-4} \mathrm{kgs}^{-2}$. This model was used to predict $u_{*,ft}$ for the median diameters of the microspheres studied in this paper. Predicting $u_{*,ft}$ provides a theoretical reference for the experimental results.

## 3 Results and Discussion

We first explain the effects of collisions on the overall detachment behavior. Then, we compare the experimental results to the model predictions for critical friction velocities. Finally, the experimental results across the range of hydrophilic to hydrophobic substrates are presented and discussed.

### 3.1 Influence of collisions on detachment

For every experiment, a collision threshold and an idealized fluid threshold were determined. Figure 3 shows the collision threshold $u_{*,ct}$ as a function of the respective idealized fluid threshold $u_{*,ift}$. The dashed line represents the idealized fluid threshold. For all data below the line, the collision threshold is smaller than the idealized fluid threshold, while for data above the line the collision threshold is higher than the idealized fluid threshold. For all microspheres $u_{*,ct}$ falls above the line for smaller $u_{*,ift}$, while it is below the line for higher $u_{*,ift}$.

In a collision, a stationary microsphere can be detached, or a rolling microsphere can be stopped. Note that $u_{*,ct}$ includes the effect of collisions. A small $u_{*,ift}$ indicates that microspheres detach and roll at low $u_*$, which results in a lower momentum than necessary for detaching stationary microspheres. Thus, the stationary microsphere stops the rolling microsphere. Hence, for a low idealized fluid threshold, the collision threshold is higher than the idealized fluid threshold. The opposite is true at high idealized fluid thresholds. Here, microspheres detach at high $u_*$ and the momentum is sufficient for detaching stationary microspheres. Thus, a rolling microsphere detaches a stationary microsphere and the collision threshold is smaller than the fluid threshold. Note that, if the microspheres moved in a hopping motion, the blocking scenario does not apply and lifting is determined solely by the fluid forces. To summarize, the data in relation to the line indicates what kind of collisions occurred. In case of data below the line, rolling microspheres detach stationary microspheres. In the opposite case, rolling microspheres are stopped by stationary microspheres.

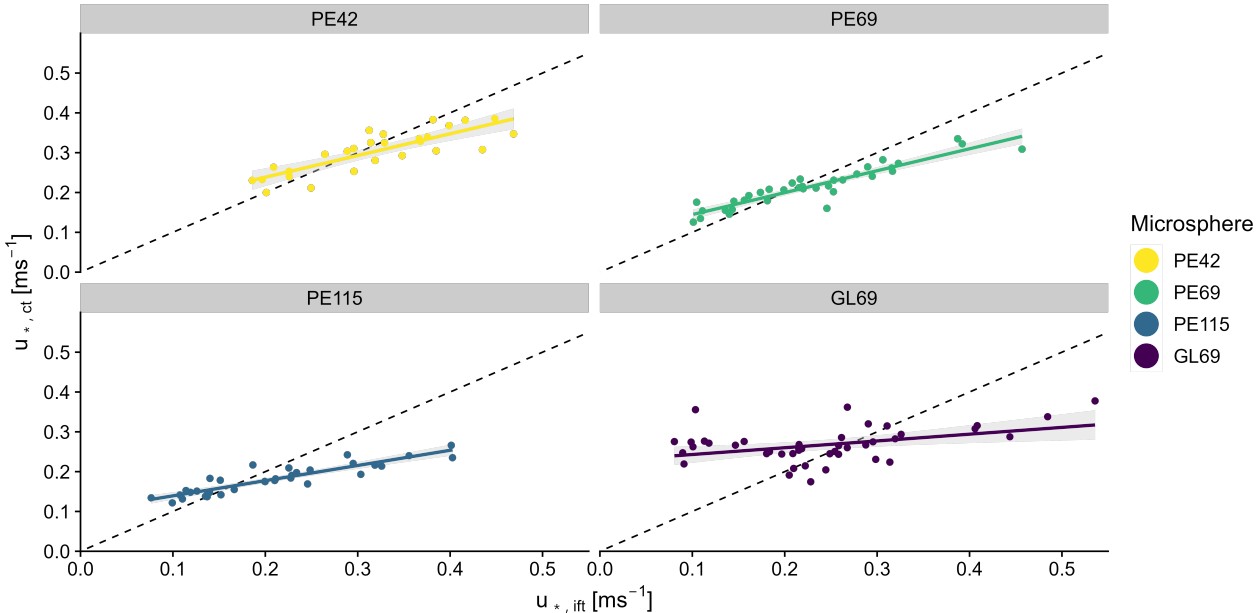

**Figure 3.** The collision threshold $u_{*,ct}$ as a function of the respective idealized fluid threshold $u_{*,ift}$. The dashed line represents the idealized fluid threshold. Results are shown for all individual experiments with polyethylene microspheres with diameters: 38-45 $\mu$m (PE42), 63-75 $\mu$m (PE69), and 106-125 $\mu$m (PE115) and borosilicate microspheres with diameters 63-75 $\mu$m (GL69).

Understanding how the collision threshold relates to the idealized fluid threshold is important when interpreting the impact of substrate hydrophobicity in Section 3.3.

## 235  3.2   Prediction and experimental results

The Shao model predicts a minimum $u_{*,ft}$ at $d = 170$ $\mu$m for PE microspheres and at $d = 108$ $\mu$m for borosilicate microspheres (see Fig. 4). For bigger or smaller microspheres $u_{*,ft}$ increases non-linearly. The model further predicts higher fluid thresholds for borosilicate microspheres than PE microspheres for particle diameters exceeding roughly $d > 20$ $\mu$m. The results presented here are all from experiments with *Substrate B*. We found the idealized fluid threshold to decrease with increasing diameter.
The larger PE69 and PE115 show similar thresholds, while GL69 detach at higher thresholds compared to PE69. Note that the results for PE69 and GL69 are graphically offset on the x-axis to improve the clarity of presentation, avoiding overlapping uncertainty bars. The dashed line indicates their true position.

The mean idealized fluid thresholds are close to their prediction by the Shao model. This finding is surprising given that the Shao model assumes stacked microspheres of identical size unlike our experimental setup. Microspheres rest upon each other,
with top particles resting in the troughs between the lower layer (see Fig. 5). The stabilizing and detaching forces are denoted by thick arrows. Their corresponding moment arms are indicated by thin arrows. When a top microsphere is detached, it pivots around point P (Fig. 5A). Similarly, a microsphere on a smooth substrate also pivots around P (Fig. 5B). Due to the different

position of P in relation to the moving microsphere, $r_d$ is bigger and $r_g$ is smaller. Hence, drag increases and lift, interparticle forces or adhesion, and the gravitational force decrease. Previous experiments showed that for the initiation of detachment, only the drag force is relevant (Kok et al., 2012). Thus, after considering the change in moment arms and using our reasoning, the idealized fluid threshold is expected to be smaller. In contrast, our experimental results nicely match the predictions and for PE42, GL69 and PE115 idealized fluid thresholds are higher than their predicted fluid thresholds. Our findings agree with the results of Ibrahim et al. (2003), who found similar idealized fluid thresholds (0.26 $ms^{-1}$) for soda lime microspheres (mean diameter 72 $\mu$m) from identically prepared substrates. We argue that idealized fluid thresholds are higher than expected due to the difference in the interparticle forces and adhesive forces between the Shao model fitted to our data and our experimental findings. Note that the Shao model was fitted to experimental data using irregular particles (Shao and Lu, 2000). Irregular particles experience less interparticle forces compared to smooth particles, due to their larger surface roughness (Cheng et al., 2002). Hence, the Shao model underestimates the interparticle forces for our experiments.

To summarize, we argue that the expected lower idealized fluid threshold based on the change in moment arms is compensated by underestimating the interparticle forces.

The Shao model predicts the point at which detaching forces overcome the stabilizing forces and a spherical particle starts to roll. We assume that the majority of particles in our experiments roll. Hence, we argue that our results can show close agreement with the prediction of the Shao model and as its assumptions also apply to our experimental setup and thus can explain our results.

The contrasting microspheres GL69 and PE69 are different in density, hydrophobicity, and surface roughness. PE69 is more hydrophobic than GL69, and PE69 had a significantly higher surface roughness compared to GL69. Hydrophobicity and surface roughness enter into the $\gamma$ parameter in the Shao model. Density is independent of $\gamma$. Note that we used an identical $\gamma = 3*10^{-4}Nm^{-1}$ for both polyethylene and borosilicate. Hence, the difference in the model predictions for the fluid thresholds was only caused by the different microsphere material densities. Here, density explains part of the difference in fluid thresholds between GL69 and PE69. As a thought experiment, we fitted the Shao model to our data giving a $\gamma$ of $3.3*10^{-4}Nm^{-1}$ and $3.7*10^{-4}Nm^{-1}$ for polyethylene and borosilicate microspheres, respectively. The difference in $\gamma$ indicates, that interparticle forces are smaller for polyethylene microspheres compared to borosilicate microspheres. We conclude that the material density and hydrophobicity are the main drivers for the lower idealized fluid threshold for polyethylene microspheres.

### 3.3 Substrate hydrophobicity and relative humidity

Figure 6 shows the detachment behavior of GL69 and PE69 in relation to $\Theta_S$. Two dashed lines indicate the predicted $u_{*,ft}$ by the Shao model for PE69 and GL69 for general reference. The mean threshold varies between 0.1 ms$^{-1}$ and 0.3 ms$^{-1}$ across all substrates. The Idealized fluid threshold and collisions thresholds are similar with a 0.05 ms$^{-1}$ difference, except for GL69 on *Substrate D* with a 0.15 ms$^{-1}$ difference. The collision threshold varies little for PE69 and GL69 with $\Theta_S$. On the contrary, the idealized fluid threshold for GL69 decreases with $\Theta_S$. The collision threshold for GL69 is higher compared to the collision threshold for PE69 independent of the substrate. However, the idealized fluid threshold for GL69 decreases with $\Theta_S$ and is smaller for $\Theta_S \geq 65°$ compared to PE69.

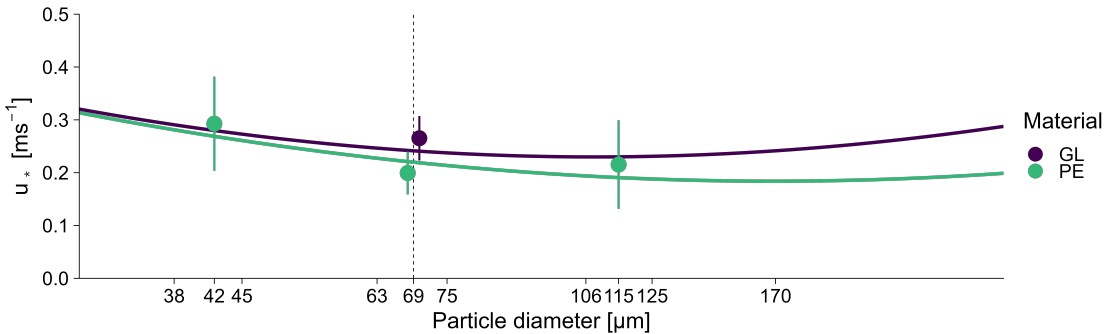

**Figure 4.** Idealized fluid thresholds as a function of microsphere diameter. Predicted fluid thresholds from the model of Shao and Lu (2000) are represented by the solid lines. Note the non-linear relationship between thresholds and the diameter. The point ranges represent the mean idealized fluid thresholds $\pm$ 1 SD of polyethylene microspheres: 38-45 $\mu$m (PE42), 63-75 $\mu$m (PE69), and 106-125 $\mu$m (PE115) and borosilicate microspheres 63-75 $\mu$m (GL69). Note that results for microsphere diameters of 69 $\mu$m were offset to the sides to improve clarity of presentation, the dashed line indicates their true position.

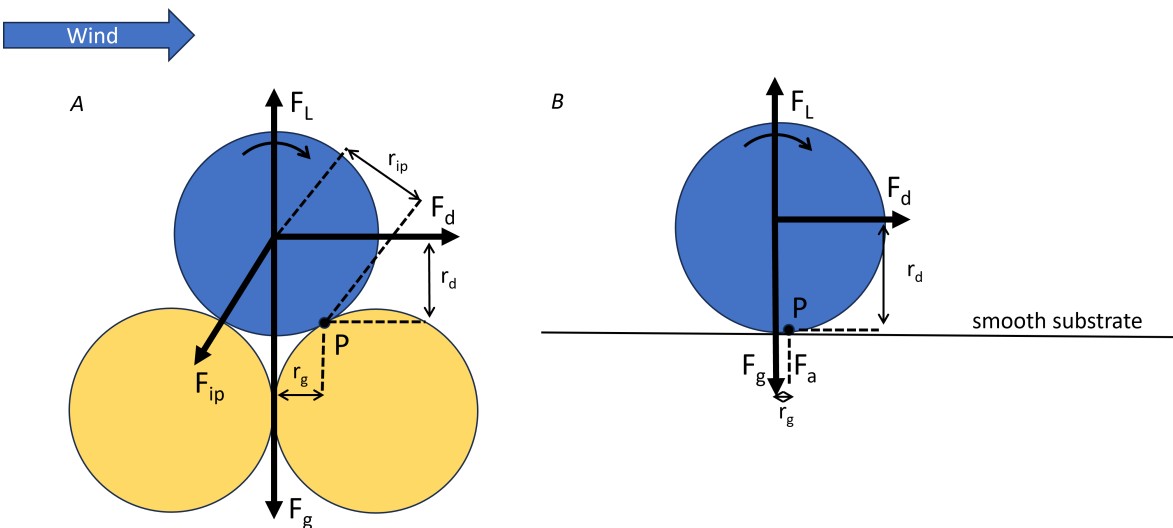

**Figure 5.** Schematic of the forces *A* acting on a particle resting on a bed of other particles (after Shao and Lu, 2000) and *B* acting on a particle resting on a smooth substrate. Thick arrows represent forces. Thin arrows represent their respective moment arms relative to the pivot point P. When the moment of the aerodynamic lift and drag forces exceeds that of the gravitational and interparticle forces, the particle will start pivoting around P in the indicated direction.

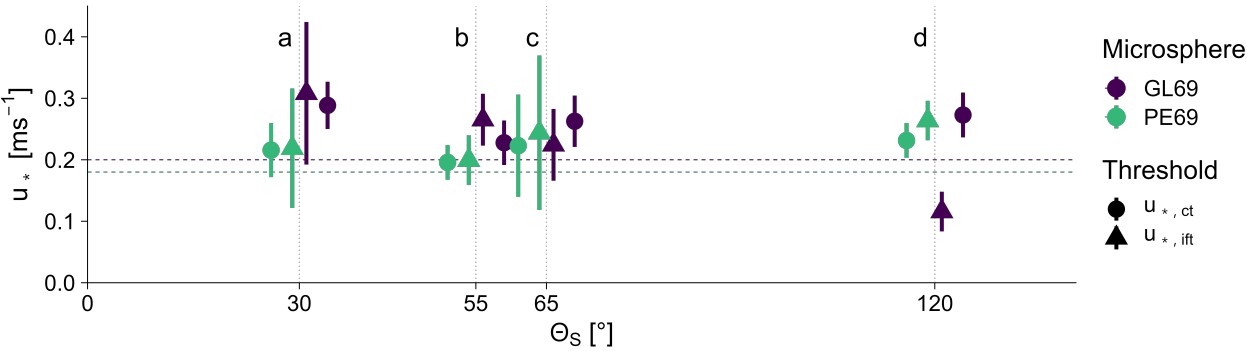

**Figure 6.** The idealized fluid threshold ($u_{*,ift}$) and collision threshold ($u_{*,ct}$) as a function of the substrate's hydrophobicity. Thresholds are contrasted for polyethylene microspheres and borosilicate microspheres on a range of hydrophilic to hydrophobic substrates. The hydrophobicity is defined as the static contact angle between the substrate and a water droplet ($\Theta_S$) using the sessile drop method. Hydrophobicity increases from small to high angles. Idealized fluid thresholds and collision thresholds are marked with triangles and dots, respectively. The point ranges represent the mean threshold $\pm 1$ SD of the respective microspheres. Indices a to d indicate the individual substrates according to Tab. 1.

We expect, in general, that the thresholds decrease with increasing hydrophobicity, as adhesion decreases and $rH_c$ increases. We expect that PE69 is less affected by $\Theta_S$ due to its hydrophobicity. Thus, we expect smaller idealized fluid thresholds for PE69, compared to GL69, due to its lower density and hydrophobic surface properties. However, the results show a more
complex pattern, which requires considering capillary forces, surface roughness, and collisions.

For GL69, the idealized fluid threshold decreases with $\Theta_S$, but the collision threshold does not. Here, collisions being effective on *Substrate A* and ineffective on *Substrate D* hide the influence of $\Theta_S$ for the collision threshold. PE69 does not exhibit dependence on $\Theta_S$ for neither the idealized fluid threshold or collision threshold.

Further, the results show that for GL69 on *Substrate A* and for PE69 on *Substrate C*, thresholds increase with rH (see. Fig.
7 and Fig. 8). No other variable changed. Therefore, we deduce that capillary forces increase idealized fluid thresholds and collision thresholds at around 30% rH for PE69 and GL69. For *Substrate D*, we expected the lowest thresholds for GL69 and PE69. However, only for GL69 the idealized fluid threshold is lowest, whereas PE69 has high idealized fluid thresholds and collision thresholds (see Fig. 9). For PE69, the high rH of 55% suggests that capillary forces increased the thresholds.

Results demonstrate that PE69 and GL69 as proxy for plastic and mineral dust, respectively, detach at $u_*$ between 0.1 to
0.3 ms$^{-1}$. In the observed range of rH, capillary forces can increase the idealized fluid threshold and collision threshold by about 0.2 ms$^{-1}$ for PE69 and GL69. Considering only the diameter and density, PE69 should detach at lower $u_*$ than GL69 according to the explored model. Further, GL69, as a hydrophilic particle, its idealized fluid threshold is dependent on $\Theta_S$, whereas PE69 is less affected by $\Theta_S$. A more detailed comparison between GL69 and PE69 regarding surface roughness and substrate hydrophobicity is masked by the influence of capillary forces. At the same rH, we would expect PE69 to detach at
smaller $u_*$, due to its lower density and hydrophobic surface, as demonstrated in Section 3.2.

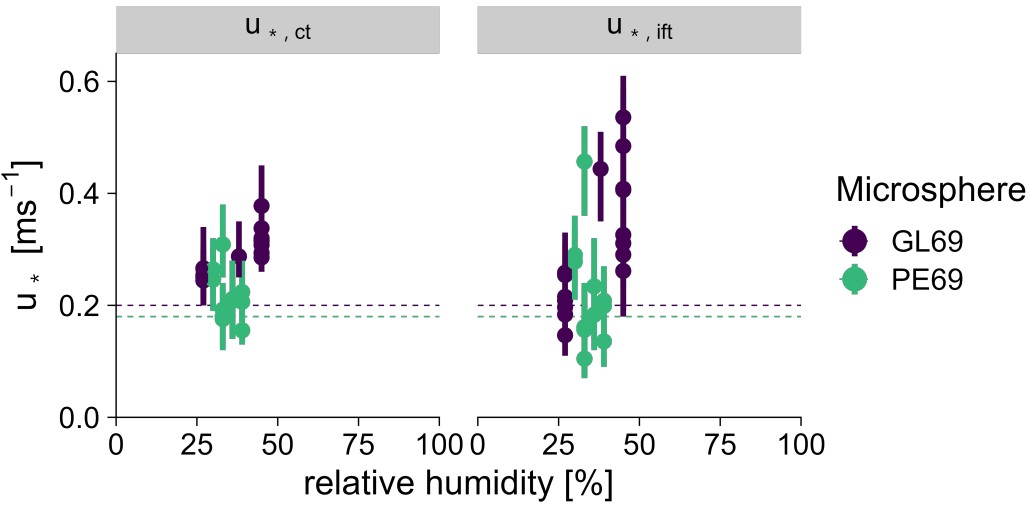

**Figure 7.** Idealized fluid thresholds ($u_{*,ift}$) and collision thresholds ($u_{*,ct}$) as a function of relative humidity (rH). Detachment is contrasted for polyethylene microspheres and borosilicate microspheres on *Substrate A*. The point ranges represent the median thresholds and their interquartile range.

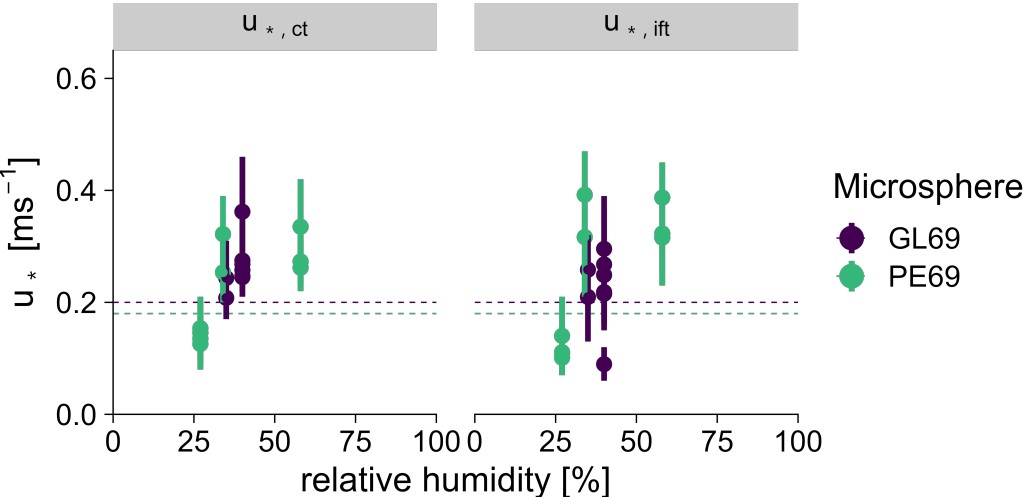

**Figure 8.** Idealized fluid thresholds ($u_{*,ift}$) and collision thresholds ($u_{*,ct}$) as a function of relative humidity (rH). Detachment is contrasted for polyethylene microspheres and borosilicate microspheres on *Substrate C*. The point ranges represent the median thresholds and their interquartile range.

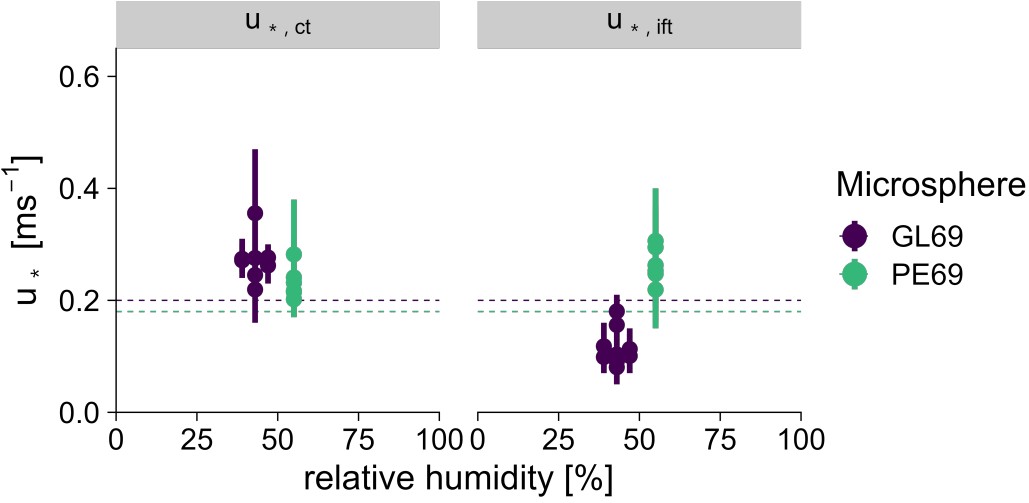

**Figure 9.** Idealized fluid thresholds ($u_{*,ift}$) and collision thresholds ($u_{*,ct}$) as a function of relative humidity (rH). Detachment is contrasted for polyethylene microspheres and borosilicate microspheres on *Substrate D*. The point ranges represent the median thresholds and their interquartile range.

### 3.4 Impact of microsphere diameter on stabilizing forces

The diameter of a microsphere determines which force dominates as the stabilizing force. For PE42, we expect that it is more dependent on $\Theta_S$ than PE115, as adhesion dominates. On the contrary, for PE115 gravity dominates, and thus we expect little variation with $\Theta_S$. The relation of the idealized fluid threshold and the collision threshold to $\Theta_S$ for PE42 and PE115 is shown in Fig.10. At the observed range of $\Theta_S$, PE42 detaches at higher thresholds. Both, the idealized fluid threshold and the collision threshold decrease with increasing $\Theta_S$. The bigger PE115 detach at lower thresholds. Further, PE115's thresholds vary little with $\Theta_S$. Both PE42 and PE114 show a high idealized fluid threshold on *Substrate C*. Differences between idealized fluid threshold and collision threshold are small, except for PE115 on *Substrate C*.

The high thresholds on *Substrate C*, found for both microsphere types, are unexpected. For experiments on *Substrate C*, we found no variable, that would explain the high thresholds by theory or by correlation. Except for *Substrate C*, the relation of thresholds and $\Theta_S$ for PE42 and PE115 fit our expectations. The bigger PE115 are less influenced by $\Theta_S$ and the idealized fluid threshold and collision threshold are close to the predicted fluid thresholds by the Shao model. The smaller PE42 are dependent on $\Theta_S$ and both thresholds decrease with increasing $\Theta_S$. Here, the variation in rH was small. Thus, there is no indication for the occurrence of capillary forces increasing neither the idealized fluid threshold nor the collision threshold (see Fig. A4).

Across substrates, PE115 behaves similar to PE69. PE115 is not sensitive to the hydrophobicity, except for Substrate C. According to the Shao model, that is reasonable, as for PE115 the gravitational force is more relevant than for PE69. On the other hand, PE42 is similar to GL69. Here, interparticle forces are dominant for PE42 and hence the thresholds decrease with hydrophobicity.

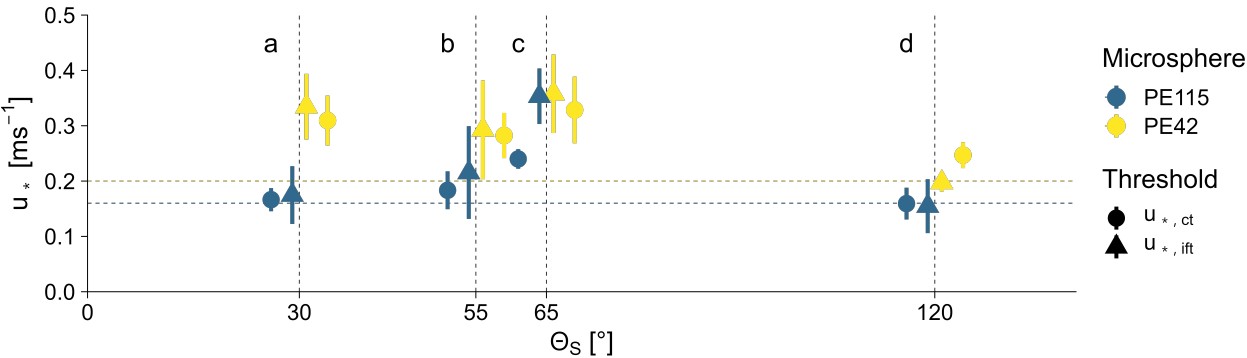

**Figure 10.** The idealized fluid threshold ($u_{*,ift}$) and collision threshold ($u_{*,ct}$) as a function of the substrate's hydrophobicity. Detachment is contrasted for polyethylene microspheres with diameters: 38-45 and 106-125 $\mu$m. Microspheres are detached from hydrophilic to hydrophobic substrates. The hydrophobicity is defined as the static contact angle between the substrate and a water droplet ($\Theta_S$) using the sessile drop method. Hydrophobicity increases from small to high angles. Idealized fluid thresholds and collisions thresholds are marked with triangles and dots, respectively. The point ranges represent the mean threshold $\pm$ 1 SD of the respective microspheres. Indices a to d indicate the individual substrates according to Tab. 1.

In addition to the results of Sect. 3.2, showed that smaller PE42 microspheres detach at similar velocities to mineral microspheres independent of their density, here the results suggest that PE42 microspheres are sensitive to the substrate's hydrophobicity.

## 4 Conclusions

Future experiments should control relative humidity, particle surface roughness and substrate surface roughness. These improvements would allow for a more precise comparison of the detachment behavior of plastic and mineral particles. Observing the detachment mechanisms would advance the understanding if plastic particles behave different from mineral particles, after they detached from a substrate.

We demonstrate that the idealized fluid threshold is useful, to examine the influence of hydrophobicity and capillary forces on detachment. Collisions can promote or mitigate detachment. Thus, one should be aware of the effects, when doing similar experiments.

The results are in good agreement with the fluid threshold predicting model by Shao and Lu (2000). At similar relative humidity polyethylene microsphere detach at smaller friction velocities compared to borosilicate microspheres of the same nominal diameter. When relative humidity increases above 30% capillary forces increase idealized fluid thresholds and collision thresholds. The idealized fluid thresholds or collisions thresholds of PE69 and PE115, did not vary with substrate hydrophobicity. The smallest polyethylene microspheres, behaved similar to borosilicate microspheres, by being sensitive to the substrate's hydrophobicity. Thresholds decreased with increasing hydrophobicity. We argue that our idealized experiments provide a use-

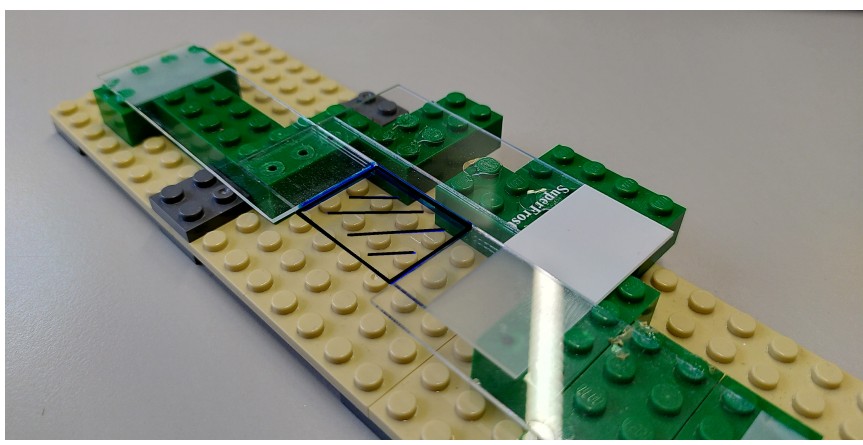

**Figure A1.** Close up on the self-made template for precise particle deposition. A substrate is placed in the template. The marked area represents the uncovered area for particle deposition.

ful analog to more complex experiments, for example using simple soils as substrate. We conclude that plastic particles are preferentially transported, as their lower density and more hydrophobic surface facilitate detachment.

*Code and data availability.* The image data of a single experiment and the code to analyze it are available at https://doi.org/10.5281/zenodo.7936729

## Appendix A

340 ### A1  Deposition template

A self-made template for microspheres deposition, ensured that all microspheres were deposited in the field of observation (see Fig. A1). When a substrate is placed in the template, the template covers the substrate, while remaining an uncovered area for particle deposition (see Fig. A2).

### A2  Substrate preparation

Glass plates (dimensions of 76×26 mm, Thermo Scientific;76×26 mm, VWR) were used as substrate material. Substrates were prepared in different fashions, thus a range from hydrophobic to hydrophilic substrates were available. The hydrophobicity of a substrate is characterized by its static contact angle with a water droplet ($\Theta_S$). Contact angle measurements were conducted using the sessile drop method (Dataphysics, Contact Angle System OCA, Filderstadt, Germany).

Before functionalization, all substrates were cleaned using a cleaning procedure popularized by the radio corporation of
America (Kern, 1990). First, the glass slide is sonicated in a 2 vol-% solution of Hellmanex III (Helma, Mühlheim, Germany) in Milli-Q water for ten minutes at 40°C and then rinsed with Milli-Q water (Merck IQ 7000, Darmstadt, Germany). It is

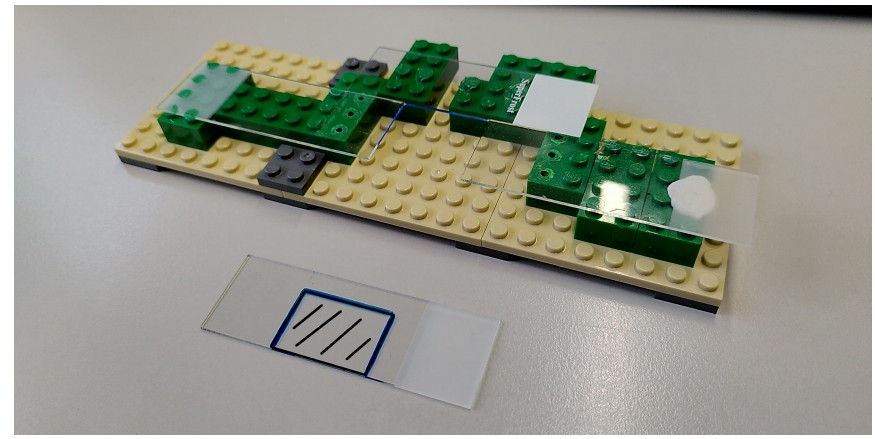

**Figure A2.** Self-made template and substrate. The marked area represents the uncovered area for particle deposition.

then sonicated in a solution of Isopropanol ($\geq$99.7%, CAS: 67-63-0, Bernd Kraft, Duisburg, Germany) and Milli-Q water in a volumetric ratio of 1:3 for ten minutes at 40°C and is again rinsed extensively with Milli-Q water. Lastly, the substrate is placed in a mixture of Milli-Q water, hydrogen peroxide (30% w/v, CAS: 7722-84-1, Fisher Chemical, Pittsburgh, Pensylvania,
USA) and ammonia (25%, CAS:7664-42-7, VWR chemicals, Radnor, Pennsylvania, USA) in a volumetric ratio of 5:1:1 for 20 minutes at 80°C, subsequently the substrate is rinsed with Milli-Q water.

Immediately after cleaning, the substrates were either used for experiments, as hydrophilic surfaces, or were further functionalized. The substrate's surface chemistry was tuned through gas phase silanisation. The substrate is placed in a desiccator onto a glass petridish, which is modified in a way, so the glass disk is elevated compared to the functionalization agent. For function-
360 alization two different silanes were used, 1H-1H-2H-2H perfluorodecyltrichlorosilane (97%, stabilized with copper, ABCR, Karlsruhe, Germany) and 3-aminopropyldimethylethoxysilane (97% ABCR, Karlsruhe, Germany). 0.5 mL of silane was placed in a petridish, under argon counter flow, and the desiccator was sealed subsequently by applying vacuum (Agilent IDP 3, Santa Clara, California, USA). The desiccator was placed in an oven at 40°C for the 1H-1H-2H-2H perfluorodecyltrichlorosilane overnight. After pressurizing, the samples were rinsed with Ethanol ($\geq$99.9%, CAS: 64-17-5, Merck, Darmstadt, Germany)
followed by Milli-Q water and immediately used for their respective experiments.

A hydrophilic substrate was prepared as described in Ibrahim et al. (2003), hereafter referred to as *Substrate B* ($\Theta_S = 55°$). Substrates were cleaned with a solution of Nitricacid 65 % (w/w) diluted with distilled water to 50 % (w/w). The substrates were submerged in the Nitricacid for 60s and washed with distilled water for 120 s. Then they were dried in a non-circulating oven at 200 °C for 1h.

**A3 Turbulence characteristics**

Boundary layer velocities were measured with a CTA at nine heights, ranging from z = 13 mm to z = 245 mm and 13 free stream velocities, ranging from 1.02 m/s to 10.87 m/s. The vertical velocity profiles showed a typical boundary-layer velocity

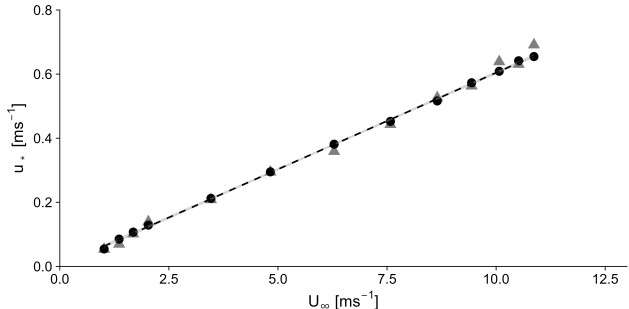

**Figure A3.** Comparison of the relationship of the free stream velocity $U_\infty$ and friction velocity $u_*$, by two approaches of determination. The friction velocity was extracted from the logarithmic velocity profile (triangles) and from covariance measurements (dots). The dashed line shows the linear regression of both approaches. $u_* = 0.06 \cdot U_\infty$ $R^2 = 0.99$

profile for a channel flow. The friction velocity and roughness length were calculated for z $\leq$ 21 mm, where the velocity profile agrees well with the logarithmic law of the wall. The roughness length ($z_0$) was calculated by extrapolating the logarithmic wind profile to the height z where $\bar{U} = 0$, giving $z_0 = 0.5\ mm$. The friction velocity ($u_*$) was computed against the free-stream velocity ($U_\infty$) in two fashions. First, it was derived from the logarithmic wind profile measured in the wind tunnel, assuming the functional form of:

$$u_{*,flux} = \kappa \frac{\delta \overline{U}}{\delta ln(z)}$$

Secondly, $u_*$ was calculated as the arithmetic mean of the directly measured density-normalized momentum flux $u_*$ using the eddy-covariance approach in the vertical profiles:

$$u_{*,EC} = \frac{1}{n} \sum_{i=1}^{n} \sqrt{-\overline{u'w'}_i}$$

Agreement among the two approaches verifies that a turbulent boundary layer has formed (see Fig. A3).

## A4  Impact of relative humidity on detachment for PE42 and PE115

Idealized fluid thresholds ($u_{*,ift}$) and collision thresholds ($u_{*,ct}$) as function of relative humidity (rH) for PE42 and PE115 on hydrophilic to hydrophobic substrates (see Fig.A4). At the observed range of $\Theta_S$, PE42 detaches at higher thresholds compared to PE115. Both thresholds for PE42 decrease with increasing hydrophobicity. For the bigger PE115, thresholds vary little with hydrophobicity. Both PE42 and PE114 show a high thresholds on *Substrate C*. Differences between idealized fluid thresholds and collision thresholds are small, except for PE115 on *Substrate C*. For *Substrate A* to *C*, detachment was measured at similar relative humidities. On *Substrate D*, thresholds are smallest overall, despite the rH being close to $50\%$.

In the observed range of rH, thresholds do not increase with rH. Thus, there is no indication for the occurrence of capillary forces. *Substrate C* shows high thresholds at similar rH found for *Substrate A* and *Substrate B*. We would expect that, at a sim-

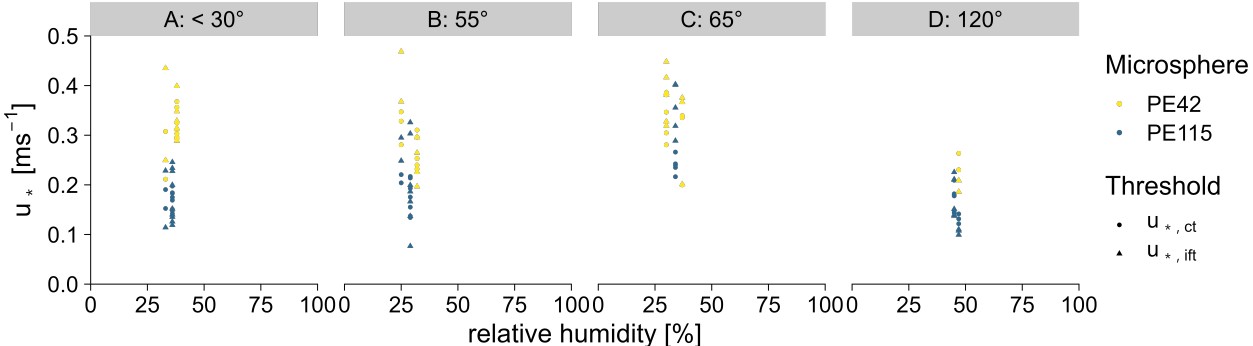

**Figure A4.** Median idealized fluid thresholds ($u_{*,ift}$) and collision thresholds ($u_{*,ct}$) as function of relative humidity for polyethylene microspheres with diameters: 38-45 $\mu$m (PE42) and 105-126 $\mu$m (PE115) on hydrophilic to hydrophobic substrates. The hydrophobicity is defined as the static contact angle between the substrate and a water droplet ($\Theta_S$) using the sessile drop method. Hydrophobicity increases from small to high angles. Indices a to d indicate the individual substrates according to Tab. 1.

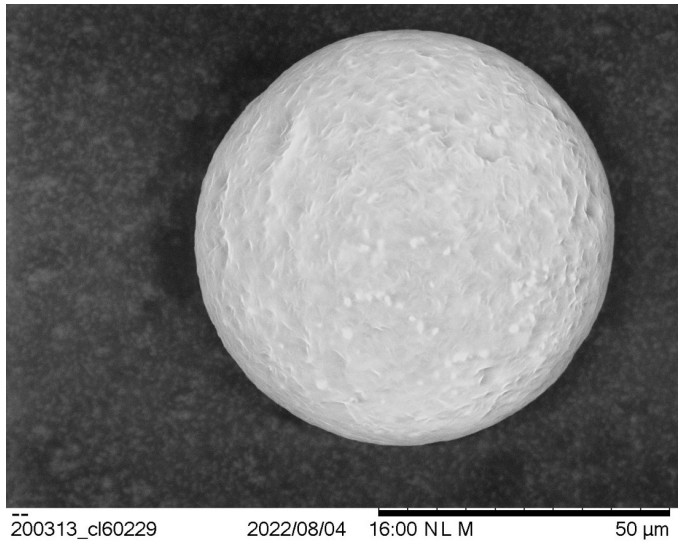

**Figure A5.** Scanning electron microscope image of a polyethylene microsphere.

ilar rH, thresholds would be lower for *Substrate C* due to the higher hydrophobicity. *Substrate D* shows the smallest thresholds for both PE42 and PE115, fitting the expectation of finding the smallest thresholds for the most hydrophobic substrate.

## A5   Scanning electron microscopy

High-resolution images of a polyethylene microsphere (see Fig. A5) and of a borosilicate microsphere (see Fig. A6) were achieved using scanning electron microscopy (Hitachi TM3030, Berkshire, UK).

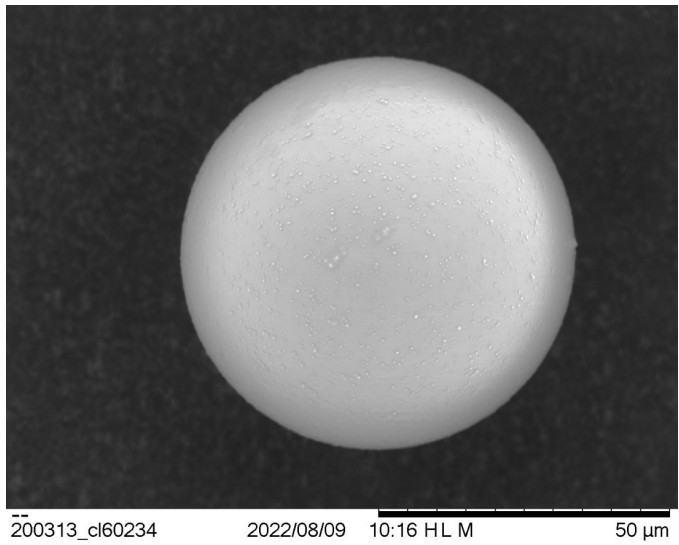

200313_cl60234          2022/08/09  10:16 H L M                    50 μm

**Figure A6.** Scanning electron microscope image of a borosilicate microsphere.

## A6 Microsphere surface roughness

A Dimension Icon AFM (Bruker Corporation Billerica, Massachusetts, USA) equipped with a NanoScope V controller was used to determine the surface roughness of PE69 and GL69. For imaging, OMCL-AC160TS cantilevers (Olympus, nominal spring constant 26 N/m, nominal resonance frequency 300 kHz) were used. The tapping mode frequency was set to 95% of the cantilevers actual resonance frequency, with an excitation amplitude of 500 mV and an amplitude setpoint of 400 mV. The AFM images were processed with NanoScope Analysis software (version 1.80, Bruker Nano Inc.). The captured data shows topographic images of the sphere caps of the investigated particles, flattened by a second-order plane fit. The images are captured by moving the cantilever over the surface with a constant setpoint. The surface topography has been determined by means of AFM operating in the so-called Tapping Mode. In this imaging the mode the AFM cantilever is driven near its first resonance frequency and a constant damping of its free amplitude, i.e. the setpoint, leads to only intermittent contact thereby preventing shear forces and tip wear. The root-mean-square roughness for PE69 and GL69 was 248.5 $\pm$ 32.2 nm and 27.7 $\pm$ 9.0 nm, respectively (see Fig. A7 and Fig. A8).

## A7 How collisions independent microsphere were determined

The following procedure outlines the process of defining windward microspheres. First, all microspheres within the field of observation are identified and added to a list of windward microspheres. For each microsphere in the list, two vectors are drawn from its center that have the same direction as the airflow. These vectors are then rotated 15° around the center of the respective microspheres, with one vector rotating clockwise and the other rotating counterclockwise. As a result, the vectors form a 30° angle, and the area within this triangle is assumed to be the space that the microsphere would pass through after

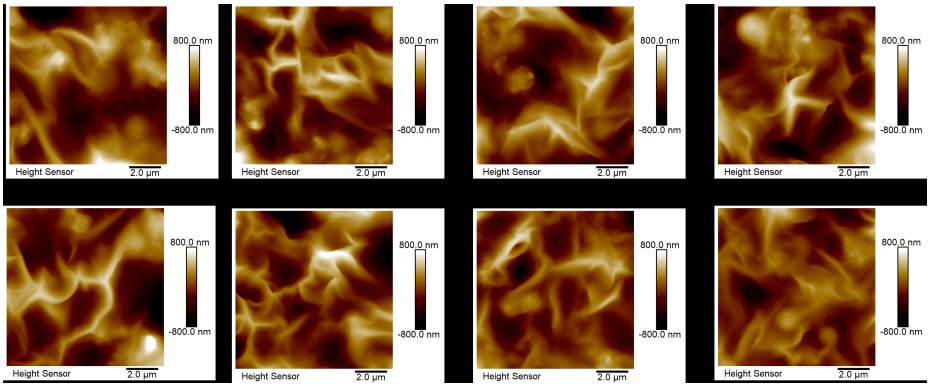

**Figure A7.** Atomic force microscopy (AFM) topography images of polyethylene microspheres. The images were captured in standard tapping mode showing the surface roughness of the sphere caps.

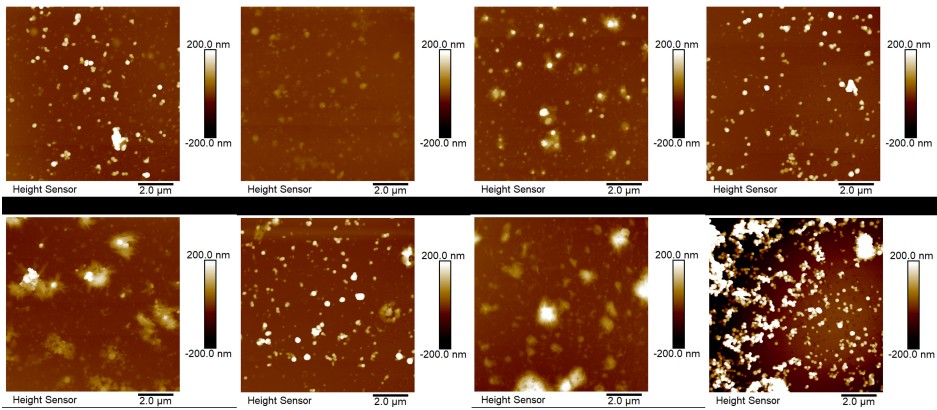

**Figure A8.** Atomic froce microscopy (AFM) topography images of borosilicate microspheres. The images were captured in standard tapping mode revealing the height profile of the sphere caps.

detaching. Consequently, any microspheres within this area are removed from the list of windward microspheres, since they are assumed to be affected by collisions with other microspheres. Once the list has been iterated through completely, it only contains windward microspheres.

## A8 Fit logistic functions, data

The data showed a logistic behavior and was well represented by a logistic function (see Fig. A9). When the increase in friction velocity achieved more than 10% and less than 30% detachment, the function fits were best. Here a range of results is presented from lowest to best fit quality.

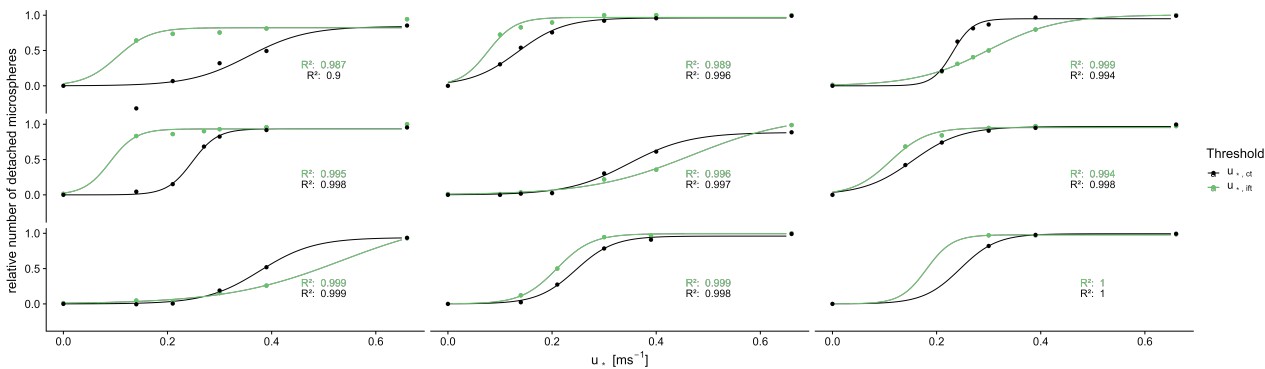

**Figure A9.** Detached microspheres as function of the friction velocity ($u_*$). Further, logistic functions fitted to the data are shown with the respective goodness of fit, represented by $R^2$. If $R^2$ is close to one, the logistic function fits to the data.

*Author contributions.* EME planned and conducted the wind tunnel experiments and wrote the manuscript; SS and IK provided and characterized the substrates and characterized the microspheres; GP, WB and CKT supervised the writing and experimental process

*Competing interests.* The authors declare that they have no competing interests.

*Disclaimer.* TEXT

*Acknowledgements.* Funded by the Deutsche Forschungsgemeinschaft (DFG, German Research Foundation) – Project Number 391977956 – SFB 1357 and 491183248. Funded by the Open Access Publication Fund of the University of Bayreuth.

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
