# Peer review of "Is transport of microplastics different from mineral particles? Idealized wind tunnel studies on polyethylene microspheres."

_EGUsphere, 2023_

## Referee Comment (RC1)

**Review:**

"Is transport of microplastics different from that of mineral dust? Results from idealized wind tunnel studies"

**Summary:**

In their manuscript, Esders and colleagues investigated the detachment process of spherical polyethylene (diameters: 42, 69, 115 um) and borosilicate (diameter: 69 um) particles from a glass slide surface using wind tunnel experiments. The study explored the influence of the substrate's hydrophobicity/hydrophilicity on the detachment process of the particles. The authors employed different cleaning and coating strategies to coat the substrates accordingly. In addition, they investigated the impact of relative humidity on the detachment process. They didn't directly control the relative humidity but rather used filtered air connected to the outside air source, with humidity levels measured continuously throughout the experiments, varying between 20 and 60 %. Furthermore, the researchers included the examination of impact-induced detachment from the spheres themselves in their data set. By comparing their experimental data with a model, the study ensures validation of their findings.

The primary aim of this explorative study is to gain first insights into the release mechanisms of microplastics in the atmosphere, starting with the investigation of one type of microplastic and comparing it to a well-studied mineral particle. The manuscript makes a valuable contribution to the field of microplastic aerosol research. Considering the significance of its findings, the work presented certainly warrants publication in ACP, after major revisions.

**GENERAL COMMENTS:**

- One major concern I have with the manuscript is the lack of discussion on the diversity of microplastics (MP). The authors exclusively studied one type of microplastic, namely polyethylene, without acknowledging the vast diversity that exists within the category of MP. Even within polyethylene (PE), there are different variations like high density PE (HDPE), linear low-density PE (LLDPE), etc. each with varying shapes, additives, and characteristics. Consequently, the title may be misleading, as the conclusions drawn are limited in scope and pertain solely to the spherical PE particles utilized in the study, not all microplastics.

  Additionally, the manuscript does not address the fluorescent nature of the PE particles used. I assume that the PE particles contain a dye that fluoresce. In this case, could the dye have an influence on the results? If yes, are the findings then comparable to other spherical PE particles that lack the fluorescent dye? I think it would be beneficial if the authors took measures to validate the material. For instance, conducting Raman or FTIR spectroscopy could provide insights into the presence and distribution of the dye or any other additives on the surface of the plastic particles used. The authors show in their

manuscript that the surface chemistry of the substrate influences the detachment process of the particles, so I assume the surface chemistry of the particles is also important, right?

**SPECIFIC COMMENTS.**

*Title:*

- The title appears to be misleading as it seemingly generalizes the study to all MP, while the research focuses solely on spherical PE particles.

*Introduction:*

- In general, the Introduction is well-composed, encompassing the current state of the art and adequately essential aspects.

- Page 2; Line 42: I miss the following citations: Materić, et al., 2022

- Page 2; Line 46, 47: There are multiple detachment processes that can occur in the environment through which MP can be released into the atmosphere. For instance, one mechanism involves MP release through bubble bursting in the ocean, which differs from the phenomena that govern detachment from soil. To provide clarity, specify and explain the detachment processes investigated by Tian et al. (2022) and Yang et al. (2022) in the second sentences of this paragraph.

*Methods:*

- All equations miss numbering.

- Page 4; Equation1: "u" is not defined in the text.

- Page 5; Line 115: Figure A5 is wrongly linked here.

- Page 5; Equation 2: "κ" is not defined in the text.

- Page 5, Line 121: The authors could perhaps briefly explain how the algorithm works to facilitate understanding of the methods.

- Page 5; Line 124: Can you add the step size?

- Page 5, Chapter 2.4: The authors do not discuss why they chose PE particles for the study. Is there any specific reason for it?

- Page 6; Table 1: Why is the contact angle for "substrate a" not described as a discrete number?

- Page 6; Line 134-136: Which method (image processing technique etc.) did the authors use to quantify the roughness of the particles? Also, the link to the Figures in the Appendix is missing.

- Page 6; Line 142: Please elaborate in more detail how the microspheres were deposited onto the substrates. Gravitational settling is not a sufficient explanation.

- Page 15, Chapter A2: I miss the discussion on how the hydrophilicity changes by the cleaning procedures. Which reactions are occurring that the surface energy changes?

*Results & Discussion:*

- Page 11; Figure 4: The caption claims that the data in the graph represents box plots. However, it is evident that it is in fact not. Please provide the correct representation or update the caption accordingly for clarity.

- In general, the authors have chosen to name the substrates alphabetically, referring to them as "substrate a," "substrate b," etc. However, this can be somewhat overwhelming when seen independently in the text. To improve readability and clarity, I would recommend using italics or capital letters to distinguish the substrates, such as "*Substrate A*," "*Substrate B*," and so on. This will make it easier for readers to identify and follow the different substrates throughout the text.

- Page 14; Chapter 3.4 (Figure 9): Chapter 3.4 misses the discussion regarding the comparison of data of PE115 & PE42 from Figure 9 to PE69, as shown in Figure 5.

*Conclusion:*

- As the dominant fraction of MP found in our environment are rather irregular-shaped fragments and fibres, wouldn't the authors think it would be a good reason to study these as well as they are environmentally more relevant than spheres?

*Appendix A:*

Page 17; Line 333: There is a space missing between the sentences: […] (see Fig.A4) At […]"

**References:**

Materić, D., Kjær, H. A., Vallelonga, P., Tison, J. L., Röckmann, T., & Holzinger, R. (2022). Nanoplastics measurements in Northern and Southern polar ice. Environmental research, 208, 112741.

Tian, X., Yang, M., Guo, Z., Chang, C., Li, J., Guo, Z., Wang, R., Li, Q., & Zou, X (2022).: Plastic mulch film induced soil microplastic enrichment and its impact on wind-blown sand and dust, Science of The Total Environment, 813, 152 490

Yang, M., Tian, X., Guo, Z., Chang, C., Li, J., Guo, Z., Li, H., Liu, R., Wang, R., Li, Q., & Zou, X. (2022): Effect of Dry Soil Aggregate Size on Microplastic Distribution and Its Implications for Microplastic Emissions Induced by Wind Erosion, Environmental Science & Technology Letters, 9, 618–624

---

## Author Response (AR1)

We would like to thank the referee for the effort and time spent reading our manuscript and posing questions and comments, which improve its relevance and clarity. Please find our responses in the following document. All comments are individually replied to.

**Review:**
"Is transport of microplastics different from that of mineral dust? Results from idealized wind tunnel studies"

**Summary:**
In their manuscript, Esders and colleagues investigated the detachment process of spherical polyethylene (diameters: 42, 69, 115 um) and borosilicate (diameter: 69 um) particles from a glass slide surface using wind tunnel experiments. The study explored the influence of the substrate's hydrophobicity/hydrophilicity on the detachment process of the particles. The authors employed different cleaning and coating strategies to coat the substrates accordingly. In addition, they investigated the impact of relative humidity on the detachment process. They didn't directly control the relative humidity but rather used filtered air connected to the outside air source, with humidity levels measured continuously throughout the experiments, varying between 20 and 60 %. Furthermore, the researchers included the examination of impact-induced detachment from the spheres themselves in their data set. By comparing their experimental data with a model, the study ensures validation of their findings.

The primary aim of this explorative study is to gain first insights into the release mechanisms of microplastics in the atmosphere, starting with the investigation of one type of microplastic and comparing it to a well-studied mineral particle. The manuscript makes a valuable contribution to the field of microplastic aerosol research. Considering the significance of its findings, the work presented certainly warrants publication in ACP, after major revisions.

**GENERAL COMMENTS:**
• One major concern I have with the manuscript is the lack of discussion on the diversity of microplastics (MP). The authors exclusively studied one type of microplastic, namely polyethylene, without acknowledging the vast diversity that exists within the category of MP. Even within polyethylene (PE), there are different variations like high density PE (HDPE), linear low-density PE (LLDPE), etc. each with varying shapes, additives, and characteristics. Consequently, the title may be misleading, as the conclusions drawn are limited in scope and pertain solely to the spherical PE particles utilized in the study, not all microplastics.

Additionally, the manuscript does not address the fluorescent nature of the PE particles used. I assume that the PE particles contain a dye that fluoresces. In this case, could the dye have an influence on the results? If yes, are the findings then comparable to other spherical PE particles that lack the fluorescent dye? I think it would be beneficial if the authors took measures to validate the material.

> *The material is 30 % fluorophore, trade secret of cospheric, but it is evenly dispersed (answer from the Company), thus we can assume, that the surface is influenced 30% fluorophore.*
>
> *Further, the leakage of dyes from particles is a process that could influence the surface chemistry. However, to the best of our knowledge there is only a number of studies that show that this process could represent a problem in mild aqueous solutions and certainly not on the time scales considered here (e.g. Kodger, Thomas E. u.a. (2017): Stable, Fluorescent Polymethylmethacrylate Particles for the Long-Term Observation of Slow*

*Colloidal Dynamics, in: Langmuir 33(25), 6382-6389. DOI: 10.1021/acs.langmuir.7b00852.). As the process takes place in air, the presence and contamination by volatile organic components will be much more dominating. However, the latter process would be present for all samples in equal manner and cannot be supressed within the experimental setup.*

For instance, conducting Raman or FTIR spectroscopy could provide insights into the presence and distribution of the dye or any other additives on the surface of the plastic particles used. The authors show in their manuscript that the surface chemistry of the substrate influences the detachment process of the particles, so I assume the surface chemistry of the particles is also important, right?

*The reviewer mentions an important point that has been admittedly not addressed sufficiently in the manuscript: The surface chemistry of both, the particles, and substrates, are of importance. However, not due to the specific functional groups but due their wetting behavior (i.e. hydrophilic vs. hydrophobic) that will control the formation of the capillary bridge between the particle and substrate (Butt, Hans-Juergen (2008): Capillary forces: Influence of roughness and heterogeneity 24(9), 4715-4721. DOI: 10.1021/la703640f.). Only in the case of silica and amino-modified slides the formation of chemical bonds in the contact area can be expected. The most promising technique to elucidate capillary and chemical forces would be the so-called colloidal probe technique, where a micro-meter sized particle of analogous composition is attached to an AFM-cantilever and the force required to remove the particle from the substrate can be measured with a force resolution in the order of a few pico-Newtons (Kappl, Michael/Butt, Hans-Jürgen (2002): The colloidal probe technique and its application to adhesion force measurements, in: Particle & Particle Systems Characterization: Measurement and Description of Particle Properties and Behavior in Powders and Other Disperse Systems 19(3), 129-143. DOI: 10.1002/1521-4117(200207)19.; Leite, FL/Herrmann, PSP (2005): Application of atomic force spectroscopy (AFS) to studies of adhesion phenomena: a review, in: Journal Of Adhesion Science And Technology 19(3-5), 365-405. DOI: 10.1163/1568561054352667.). However, this technique is unfortunately out of the scope for this study.*

*Additional Comment:*

*We changed the plot showing the ratio of the idealized fluid threshold (previous CIMs) and collision threshold (previous CDMs) as a function of the idealized fluid threshold. Plotting any mathematical operation of A and B as a function of A leads to spurious correlation. Thus, we changed to plotting the collision thresholds as a function of the idealized fluid thresholds. And describe the difference between them, as the result of collisions. See Section 3.1.*

**SPECIFIC COMMENTS.**

*Title:*
• The title appears to be misleading as it seemingly generalizes the study to all MP, while the research focuses solely on spherical PE particles.

> *We changed the title of the manuscript to: Is transport of microplastics different from mineral particles? Idealized wind tunnel studies on polyethylene microspheres. Thus, it should be clear, that we focus on one prominent representative of microplastics.*

*Introduction:*
• In general, the Introduction is well-composed, encompassing the current state of the art and adequately essential aspects.

> *Thanks for the positive feedback.*

• Page 2; Line 42: I miss the following citations: Materić, et al., 2022

> *We added the citation. Line 46*

• Page 2; Line 46, 47: There are multiple detachment processes that can occur in the environment through which MP can be released into the atmosphere. For instance, one mechanism involves MP release through bubble bursting in the ocean, which differs from the phenomena that govern detachment from soil. To provide clarity, specify and explain the detachment processes investigated by Tian et al. (2022) and Yang et al. (2022) in the second sentences of this paragraph.

> *We added a description of the emission mechanisms.*

*Methods:*
• All equations miss numbering.

> *We added numbering to all equations.*

• Page 4; Equation1: "u" is not defined in the text.

> *We added a definition of u.*

• Page 5; Line 115: Figure A5 is wrongly linked here.

> *We corrected the link.*

• Page 5; Equation 2: "κ" is not defined in the text.

> *We added a definition of κ.*

• Page 5, Line 121: The authors could perhaps briefly explain how the algorithm works to facilitate understanding of the methods.

> *We added a brief explanation.*

• Page 5; Line 124: Can you add the step size?

*The step size varied, as the adhesive forces varied, due to size, material and air humidity. Thus, we choose to describe the step-size with the intention of a similar percentage of detached microspheres for each step.*

• Page 5, Chapter 2.4: The authors do not discuss why they chose PE particles for the study. Is there any specific reason for it?

*We used these specific PE microspheres, as PE is one of the commonly used plastics, it was readily available as fluorescent microspheres, that are necessary for a robust detection, and borosilicate microspheres with the same diameter were available as reference material.*

• Page 6; Table 1: Why is the contact angle for "substrate a" not described as a discrete number?

*For substrate a no discrete contact angle could be determined, because water completely wets the substrate and no droplet forms, indicating a highly hydrophilic substrate.*

• Page 6; Line 134-136: Which method (image processing technique etc.) did the authors use to quantify the roughness of the particles? Also, the link to the Figures in the Appendix is missing.

*We added the link to the appendix. The method is described in lines 395-406*

• Page 6; Line 142: Please elaborate in more detail how the microspheres were deposited onto the substrates. Gravitational settling is not a sufficient explanation.

*We added further explanation, on how the microspheres were deposited. See lines 163-165.*

• Page 15, Chapter A2: I miss the discussion on how the hydrophilicity changes by the cleaning procedures. Which reactions are occurring that the surface energy changes?

*The cleaning procedure applied is well known in the field of surface science and goes back to Kern and coworkers.[ Kern, W/Puotinen, D A (1970): Cleaning Solutions Based on Hydrogen Peroxide for Use in Silicon Semiconductor Technology, in: Rca Review 31(2), 187.] It is known under the name 'RCA-cleaning', after the company where it has been developed. Here, only the second part of the cleaning procedure has been utilized, which comprises a controlled oxidation of the surface to form Si-OH groups at the surface.*

*Results & Discussion:*
• Page 11; Figure 4: The caption claims that the data in the graph represents box plots. However, it is evident that it is in fact not. Please provide the correct representation or update the caption accordingly for clarity.

*We corrected the caption.*

• In general, the authors have chosen to name the substrates alphabetically, referring to them as "substrate a," "substrate b," etc. However, this can be somewhat overwhelming when seen independently in the text. To improve readability and clarity, I would recommend using italics or

capital letters to distinguish the substrates, such as "*Substrate A*," "*Substrate B*," and so on. This will make it easier for readers to identify and follow the different substrates throughout the text.

*We changed to the proposed representation.*

• Page 14; Chapter 3.4 (Figure 9): Chapter 3.4 misses the discussion regarding the comparison of data of PE115 & PE42 from Figure 9 to PE69, as shown in Figure 5.

*We added discussion regarding PE115 and PE42 in the context with PE69 and GL69. See lines 315-318.*

*Conclusion:*
• As the dominant fraction of MP found in our environment are rather irregular-shaped fragments and fibres, wouldn't the authors think it would be a good reason to study these as well as they are environmentally more relevant than spheres?

*We agree that studying MP geometries that occur more often in the environment are important. Studying fibers and irregularly shaped particles should be a priority in the future. However, this manuscript presents a first step in studying the atmospheric transport of MP, with a focus on simple geometry and idealized substrates. In the future we will use the insights from the current manuscript to work on more complicated particles, substrates, and emission mechanisms in a controllable environment.*

*Appendix A:*
Page 17; Line 333: There is a space missing between the sentences: […] (see Fig.A4) At […]"

*We corrected the spacing.*

**References:**
Materić, D., Kjær, H. A., Vallelonga, P., Tison, J. L., Röckmann, T., & Holzinger, R. (2022). Nanoplastics measurements in Northern and Southern polar ice. Environmental research, 208, 112741.

Tian, X., Yang, M., Guo, Z., Chang, C., Li, J., Guo, Z., Wang, R., Li, Q., & Zou, X (2022).: Plastic mulch film induced soil microplastic enrichment and its impact on wind-blown sand and dust, Science of The Total Environment, 813, 152 490

Yang, M., Tian, X., Guo, Z., Chang, C., Li, J., Guo, Z., Li, H., Liu, R., Wang, R., Li, Q., & Zou, X. (2022): Effect of Dry Soil Aggregate Size on Microplastic Distribution and Its Implications for Microplastic Emissions Induced by Wind Erosion, Environmental Science & Technology Letters, 9, 618–624

*We would like to thank the referee for the effort and time spent reading our manuscript and posing questions and comments, which improve its relevance and clarity. Please find our responses in the following document. All comments are individually replied to.*

Enders et al. investigate the wind movement of microplastics in the wind tunnel using different sized polyethylene (PE) and borosilicate (GL) microspheres, the latter as a reference mimicking dust (or maybe better sand) particles. The study is very timely: microplastics and their atmospheric transport have become a focus of research, yet the mechanism of their entrainment from a land/soil surface into the atmosphere is not yet well understood. By comparing the aeolian transport of microplastic with that of mineral dust – which, even though there are still important gaps in our understanding of dust emission as well, can build on a body of research of several decades – the authors take an important step in trying to advance the process understanding of microplastic emission.

The manuscript is well written overall and most results are well explained. However, I do see some conceptual and experimental design aspects, which need clarification. I detail those together with some other aspects in the following. I hope that my comments will help to further improve the study and make it even more relevant and insightful.

**Major comments:**

1. Rather than placing the test microspheres, either PE or GL, on e.g. a soil substrate, the authors use a glass plate as a substrate. A glass plate, different to a "natural" surface (meaning a surface in the outside environment), is very smooth. While I understand that this eases implementation of the experiment, it has two important disadvantages: i) the microspheres will roll off the (very, very small) field of observation rather than lift off; ii) in the case of a soil surface, e.g. a desert, no interaction between microplastics and soil particles (sand/dust) can be considered. Point ii) may not be relevant for other surfaces, such as asphalt. Point i), however, is very important in my opinion, because the mechanism relevant for long-term atmospheric transport of microplastics is suspension and not creep. The authors touch upon this topic somewhat when discussing the expression for wind erosion threshold by Shao and Lu (2000) [I would like to add that this is not a wind erosion model, contrary to what is stated in the text], which is primarily addressing saltation (l. 212 – 225). I therefore wonder why the authors chose this experimental design, which does not address the target emission mechanism. Related to that, the primary emission mechanism of mineral dust is saltation bombardment (e.g. Shao et al., 1993; Shao, 2008; Kok et al., 2012). Is this also expected to be the main emission mechanism for microplastics (note that it requires microplastic particles to impact upon, hence for plastic/plastic collisions a relatively large abundance of microplastics on a surface is needed; less so for sand/plastic collisions) or – due to the lower density – maybe aerodynamic entrainment (e.g. Klose et al., 2014)? And what is the effect of the often very non-spherical shape of microplastics on their emission? Why did the authors choose to study the movement threshold of two spherical particle types with main difference being their density (which has been investigated in previous studies; e.g. Corn and Stein, 1965; Iversen and White, 1982; etc.) over the other questions? I believe this requires a more detailed conceptual discussion and justification. Also, previous experimental studies that investigated the wind transport of different-density particles should be discussed more.

*We agree, that many further steps need to be taken, to gain further insights in the atmospheric transport of microplastics. We here present a first step in the direction and start with an idealized approach to gain fundamental insights in the difference between plastic and mineral microspheres.*

2. I find the terminology of individual critical friction velocity, critical friction velocity, and threshold friction velocity unfortunate, given that it does not seem to agree with that used in the wind erosion / aeolian research community. Also, the benefit of defining the critical friction velocity as that for 25% detached particles and threshold friction velocity as 50% detached particles is not clear to me. Please consider using comparable terminology. Note also that the aeolian research community differentiates between the fluid and impact thresholds, the latter accounting for particle collisions (e.g. Kok et al., 2012) and that threshold friction velocity is primarily used in the context of saltation.

> *We changed the terminology thus it also refers to the detaching forces. Thus, we introduced an idealized fluid threshold, which applies when only the fluid forces cause the detachment. We named it idealized as it is determined from our idealized experiments, where a microsphere detaches from a smooth substrate and only experiences adhesive and gravitational forces as stabilizing forces. Further, we introduced the collision threshold. It represents the threshold for collision influenced microspheres.*

3. Please add more detail on the experiments, e.g. how many experiments have been conducted in total, how many replicates for each substrate/microsphere type, etc. In line 145 it is only mentioned "over 30 runs". Those should be defined exactly.

> *We added a table stating exact numbers of replicates, regarding every microsphere substrate combination. See top of page 7.*

4. It is noted that the substrate is placed on top of the roughness elements, which are ~1 cm (?) high and that the aerodynamic roughness length is 0.5 mm. From my perspective this means that the substrate and particles are not actually at the surface, but elevated? How does this impact the accuracy of the derived values of friction velocity for the height of the substrate?

> *It is correct that the substrates are elevated. We chose this experimental setup due to the following factors. We aimed for a high surface roughness in the wind tunnel, to achieve friction velocities up to 0.65 m/s at a maximum free-stream velocity of 11m/s. Surface roughness was on the scale of the substrate itself, thus it was possible to position it at the top of the roughness elements, the bottom or somewhere in between. We positioned the substrate at the top of the roughness elements, thus it would be above the roughness layer of the roughness elements, but inside the newly formed equilibrium layer. The elevation of the substrate does not impact the accuracy of the derived friction velocity. In a boundary layer the friction velocity is height independent.*

5. Generally, as commented upon above, the comparison with mineral dust throughout the manuscript is somehow problematic as only creep is considered in the experiments and not suspension, even though I understand that the ultimate ambition is to compare with mineral dust transport. Also, the size of the used mineral dust analogue, GL69, is not in the typical dust size range, but would be giant mineral dust or sand.

> *Comparing polyethylene and borosilicate microspheres is a first step towards a better understanding of microplastic transport via the atmosphere. The Shao model is the latest reiteration of the scheme introduced by Bagnold in 1941. It is a simple model predicting fluid thresholds. Ultimately it predicts at what friction velocity a particle starts pivoting around its pivot. Hence, our experiments observe a similar mechanism. That is why, we argue, that our results can be compared to the Shao model and give insight into detachment mechanisms for borosilicate and polyethylene microspheres in general.*
>
> *Additional Comment:*
>
> *We changed the plot showing the ratio of the idealized fluid threshold (previous CIMs) and collision threshold (previous CDMs) as a function of the idealized fluid threshold. Plotting any mathematical operation of A and B as a function of A leads to spurious correlation. Thus, we changed to plotting the collision thresholds as a function of the idealized fluid thresholds. And describe the difference between them, as the result of collisions. See Section 3.1.*

**Comments on the introduction**

**Technical comments:**

- Title: Both the words "transport" and "mineral dust" are misleading, because they indicate long-term transport, whereas the processes considered experimentally in the present study are (short-term) creep of sand-sized particles.

  > *We changed the title of the manuscript to: Is transport of microplastics different from mineral particles? Idealized wind tunnel studies on polyethylene microspheres.*
  >
  > *We specified what kind of particles were examined. Further, the term "idealized" points to the reduced complexity of the experiment and that another first approach to the question is presented. We would argue that detachment is the very first step of transport. Hence, we keep the word "transport", to make sure the direction of the study is obvious.*

- L 1 add "can" after "Atmospheric transport"

  > *We added "can" after "Atmospheric transport".*

- L 2 Knowing all the challenges associated with mineral dust transport (e.g. related with particle shape), I would very much argue that long-range transport of microplastics is not at all well-known. The author in fact state the same in line 44.

> *We deleted the statement about long-range transport from the text.*

- L 3 contrasted them

  *Corrected*

- L 10-11 The statement that collisions can both enhance and mitigate detachment needs more explanation, otherwise it is not understandable without context.

  *We added more explanation about the mechanisms in collisions leading to the enhancement or mitigation on detachment. See lines 9-10.*

- L 14 borosilicate instead of mineral microspheres

  *Corrected*

  L 15 fitting to the prediction of the wind erosion threshold friction velocity (adapt to the terminology used in the paper; cf. comment (2)

  *We changed the terminology.*

- L 19-20 The conclusion made here is not justified by the paper's results. Suggest removing.

  *We adjusted the conclusion. See lines 21-22.*

- L 56 "driving the emission"

  *We changed the wording accordingly*

- L 65-66 I assume that rHc refers to the relative humidity in air. With "water accumulating", do you mean that the water vapor condenses on the microspheres and substrates?

  *We assume that, at a critical relative humidity water condenses between a microsphere and substrate. The condensed water forms a capillary meniscus, that causes an attractive force.*

- L 76 define subscript i (if you stick to this terminology)

  *We changed the terminology and took out subscript i.*

- L 78-79 The behavior of the microspheres would likely be quite different if they were "embedded" in a rough substrate, rather than resting on top of a smooth plate.

  *We agree and added a note in the introduction. See lines 87-88.*

- L 83 collisions can lead (compare next sentence)

*We added "can".*

- L 86-88 check grammar for i) and ii) (To what extent do collisions influence …; Do the findings support …)

    *We corrected the grammar.*

- L 88 Doesn't the finding of a preferential support of microplastics explicitly relate to the concomitant entrainment of both sand/dust and microplastics (e.g. Bullard et el., 2021) rather than the density-dependent entrainment (here movement) friction velocity threshold? For the latter, as mentioned before, I strongly recommend to discuss more of the previous study on this topic.

    *We examined the effects of size, surface properties and relative humidity on detachment. Thus, we provide insight on how strong the attraction between polyethylene or borosilicate microspheres are to substrates with small to high hydrophobicity. We argue that a loosely bound particle is easier transported, also if saltation is the emission mechanism, as the attraction forces that must be overcome by the sand particle is smaller.*

**Comments on Methods**

- L 108 Where/which height is the free-stream velocity measured?

    *We added an explanation, where the free-stream velocity was measured. See lines 117-118.*

- L 112 10 s appear like a very long interval to track particle movement. Was a higher temporal frequency not possible or was this interval chosen on purpose? Please comment on this choice.

    *With our camera system 0.125 Hz was the highest temporal frequency feasible. Frame rates in the magnitude of 4kHz are necessary to investigate the detachment mechanisms of single microspheres ( Kassab et al. 2013). Thus, at observing the detachment mechanisms itself was out of the question, we reduced the temporal frequency to 0.1 Hz to reduce the amount of memory needed to store all the data.*

- Sec 2.2, line 1: Should the Fig. reference be to Fig. A3?

    *We corrected the reference.*

- L 120 Related to the previous comment, by counting only the remaining number of particles, it is impossible to know the detachment mechanism. Was the movement of the microspheres not observed/recorded in some way?

    *Unfortunately, with the temporal resolution it was not possible to observe the detachment mechanism. Additional measurement techniques to capture the detachment mechanism were out of the scope of the study.*

- Table 1: Please add substrate material

  *We added the material to the table.*

- Sec 2.5, line before second equation (the line numbering seems off here): The authors mention that a logistic function was fit to the experimental results, but – unless I overlooked it – it is never shown how well this fit matches the results. Please give more information so that the reader can understand the accuracy of the fit. On another note, it is not clear to me why A/2 in the equation is replaced by m, but I leave this to authors' preference.

  *We added a comprehensive plot showing how the results fitted to a logistic function. See Fig A9. Further, we simplified the equation by taking out the variable m, just as suggested.*

- L 153 Do I understand correctly that u*th is defined as the friction velocity at which – in one entire experiment – 50% of all particles have detached / are remaining? It is not determined as detachment of 50% of all microsphere at one given time/friction velocity interval, right? The wording is not entirely clear. Please also comment on how this definition relates to the threshold friction velocity used in wind erosion research.

  *We rephrased the sentence, such that it is clearer that we define the threshold as the friction velocity at which 50% of all particles of a single experiment detached. See lines 193-195.*

  *We added explanation of how we compare the now defined idealized fluid threshold to the fluid threshold predicted by the Shao model. See lines 243-258.*

  On another note, in line 182, the critical friction velocity is defined as that friction velocity at which 25% of the particles detached. Why are two different particle number thresholds (25% and 50%) used? The purpose is not clear to me.

  *We changed the terminology and now use only the idealized fluid threshold to compare with the fluid threshold of the Shao model. It is defined as the friction velocity at which 50% of the most windward particles detached.*

- L 155-157 See also Shao and Klose (2016, https://doi.org/10.1016/j.aeolia.2016.08.004)

  *The reference advanced our understanding of the stochastic nature of cohesive forces. It lead us, to define only an idealized fluid threshold and a collision threshold. We know interpret the spread in detachment as the result of the stochastic nature of adhesion and turbulence. See lines 196-200.*

- Sec 2.7, line 1: The critical friction velocity is here defined as for "multiple microspheres", which I interpret as for a population of microspheres. I find this somehow misleading though and would argue that u*c, especially in the context of the estimate by Shao and Lu (2000) is the average critical (or threshold) friction velocity.

*We changed the terminology. Now there is only thresholds defined, depending on which detaching forces are involved (the fluid, or the fluid plus collisions) and no differentiation between single and multiple microspheres.*

- Sec 2.7 lines 3-4: It is important to say, as also discussed by the authors later on, that Shao and Lu (2000) assume that particles are resting on top of each other, contrary to the monolayer of particles considered here. I suggest to rephrase the sentence "We assume that a glass plate … represents a simplified soil" accordingly.

  *We rephrased the sentence. See lines 206-207.*

- L 178-179 It is important to note here that both Ravi et al. and McKenna Neuman et al. determined this threshold for saltation as measured by, e.g. impact sensors.

  *We deleted the paragraph, as we now compared the predicted fluid thresholds to idealized fluid thresholds.*

**Comments on Results and Discussion**

- L 195 A smaller u*th for CIM at low u*th seems indeed only applicable for rolling and sliding motions, because then the fluid motion can be inhibited by blocking stationary particles. It does not apply to a hopping motion, because then the blocking does not apply and lifting is determined by the fluid forces alone.

  *We added the note. See lines 229-230.*

- L 204 d < 100 um (70 um) < d is not a mathematically correct formulation, because d cannot at the same time be smaller and larger than a reference. Please revise.

  *We rephrased the description.*

- L 209 It is quite surprising that a lower threshold/critical friction velocity than estimated by Shao and Lu (2000) is only found for two types of microspheres, even though a lower threshold is expected for rolling compared to lifting. How do you explain that?

  *We now defined an idealized fluid threshold and compare it with the predicted fluid thresholds by the Shao model. Experiment and prediction are close to each other, and we explain that by a change in moment arms, that leads to an expected lower idealized fluid threshold, but also an increase in adhesion, that compensates for that. See lines 243-258.*

- L 212 I think a better formulation than "overpredicts u*c compared to our observations" would be "it is conceivable that we obtain lower u*c from our experiments than predicted by…". "Overpredict" creates the notion that the estimate by Shao and Lu (2000) is wrong, whereas the difference here is at a first instance due to a different assumed setting.

  *We corrected the wording.*

- L 228 It seems that the effect of nano-scale surface roughness has received attention particularly in the context of capillary forces (e.g. Rabinovich et al, 2002; Kim et al., 2016). It is correct though that it is not considered in the Shao and Lu model. I think this topic is very interesting, however, I would argue that it is of second order importance compared to differences between the shape of "natural" compared to spherical particles, which may be even more important for microplastics compared to sand/dust.

  *We agree that, the shape of particles in the observed size range is more important than nanoscale roughness, when it comes to emission potentials.*

- L 230-231 The last sentence is not clear. Suggest revising.

  *We revised the paragraph and the sentence was deleted.*

- L 246-248 It seems that the variability of u*th with rH is quite diverse. How do the authors explain that u*th is not always increasing with rH? Is there any further insight from the experiments on the cause of this? I believe a somewhat more detailed discussion would be insightful.

  *We agree, but we have no further data than the relative air humidity to explain the increase in idealized fluid thresholds.*

- Figures: Please explain how the error range has been calculated.

  *We added the explanation.*

- Please list references in a systematic order, e.g. chronological (recommended) or alphabetical.

  *We listed the references now in a chronological order.*

- Fig. 5 Please add explanation for the circles and triangles in all figures / figure captions.

  *We added explanation for the circles and triangles in Fig.5 and Fig.9*

- L 262 decreases with increasing theta_s

  *We added the missing "increasing".*

- L 268-269 Would an indication for the occurrence of capillary forces be meant to be due to a variation in rH? The context of this statement is not quite clear.

  *We rephrased the sentence. See lines 313-314.*

- L 270 which showed instead of showing

  *We corrected the wording.*

- L 276 particle surface roughness

  *We corrected the wording.*

**Comments on the conclusion**

- L 274 – 277 Please see my initial comments on aspects which from my perspective are more pressing and more tailored to advancing understanding of the entrainment of microplastics.

  *We like to keep the focus of understanding the transport mechanism using idealized approaches. We added the note, that observing the detachment meachnism would be important to advance the understanding of the entrainment of microplastics.*

- L 278-280 I find that this passage does not summarize the results very precisely. I strongly recommend to be more specific on the achievements made in this study, in particular define what is meant by detachment behavior.

  *We added a more detailed summary of our achievements. See lines 330-337.*

- L 282 I would not agree that the agreement confirms that a glass plate equipped with a monolayer of microspheres represents a simplified soil. Please see my earlier similar comment and rephrase the sentence accordingly.

  *We rephrased the sentence. We argue that the idealized approach is a useful analog to simple soils. See lines 335-336.*

- L 283-285 As mentioned in my comment 1, mineral dust is entrained either through saltation as intermediate mechanism or directly aerodynamically by wind. What is studied here for microplastics is the initiation of creep, which does not yet allow to conclude that the behavior of microplastics is similar to that of dust, in particular considering that saltation (to entrain dust) is driven by sand grains which are (not exactly, but somewhat) spherical. This may not apply in the same way to microplastics, or does it? The smaller density of microplastics compared to dust may certainly play a role, but without knowing the emission mechanism, its roll cannot be conclusively determined (e.g. spherical microplastics may entrain smaller particles less efficiently through saltation than sand does, do to the lower density).

  *We rephrased the sentence. Still, we hypothesize that as the detachment of polyethylene is facilitated by its lower density and hydrophobicity, and detachment is the very first step to rolling, creeping or entrainment, plastic particles will be preferentially transported by wind. See lines 336-337*

- L 285 Similarly as in the abstract, the last sentence is not supported by the study results as suspension is not (yet) investigated.

  *We added the hypothesis, that plastic particles are preferentially transported, due to their lower density and hydrophobicity. See lines 336-337.*

**Comments on the Appendix**

- A3: should "(see A3)" be "(see Fig. A3)"?; also lines 329-330 are the same as in the main text.

  *We corrected the reference and took out the redundant text.*

- A4: I suggest to place all four panels side-by-side such that the magnitudes can be compared more easily. It seems that there is enough space for that.

  *We adjusted the plot.*

- There is no reference to Appendices A5 to A7 in the main text.

  *We added references in the main text.*

- L 333 There is no pseudocode in this section. Suggest calling it procedure instead.

  *We changed the wording to "procedure".*

- A6: This section is not understandable for someone who has not used this technique before. Please add some more detail, also to Figs. A7 and A8

  *We revised the wording and added further explanation of the technique. See lines 396-406.*

---

## Referee Report (RR1)

**Review_R2:**

"Is transport of microplastics different from mineral particles? Idealized wind tunnel studies on polyethylene microspheres."

The authors have successfully addressed the issues I raised and revised the manuscript accordingly, leading me to conclude that the manuscript is now suitable for publication, after technical corrections.

**TECHNICAL CORRECTIONS:**

- Title: I appreciate the updated title with the inclusion of 'polyethylene' for specificity. However, I suggest removing the dot at the end, as it seems unconventional.

- Tables: The caption of tables should be above the table, not below.

- Appendix: The formulas in the appendix lack numbering.